# Graph Unitary Message Passing

**Haiquan Qiu**[1]                                                                          *qhq22@mails.tsinghua.edu.cn*
**Quanming Yao**[1,2,3]                                                                    *qyaoaa@tsinghua.edu.cn*
[1] *Department of Electrical Engineering, Tsinghua University*
[2] *Beijing National Research Center for Information Science and Technology*
[3] *State Key laboratory of Space Network and Communications*

**Reviewed on OpenReview:** *https://openreview.net/forum?id=dvNMDkSBIA*

## Abstract

Unitarity is a useful principle for stabilizing deep neural networks, but in graph neural networks (GNNs) instability is induced not only by learnable parameters but also by the graph propagation operator. Motivated by this distinction, we propose Graph Unitary Message Passing (GUMP), a message-passing framework that uses a unitary propagation operator on a transformed graph to avoid graph-induced exponential decay under repeated propagation. GUMP combines (i) a graph transformation that maps an input graph to an Eulerian line-graph construction admitting unitary adjacency matrices, and (ii) a practical unitary projection procedure based on Newton-Schulz iteration. Theoretical analysis clarifies that, under standard analysis assumptions, unitary propagation keeps the graph-propagation term depth-stable, while vanilla normalized propagation exhibits exponential decay in its non-trivial spectral components. Across synthetic long-range tasks, TUDataset benchmarks, and LRGB datasets, GUMP improves over vanilla message passing and achieves competitive or superior performance against strong baselines. Code is available at `https://github.com/ucker/gump_code`.

## 1 Introduction

Unitarity has become an important principle for stabilizing deep neural networks and preventing gradient pathologies. Prior work has applied it through orthogonal or unitary initialization (Saxe et al., 2013; Arjovsky et al., 2016; Orvieto et al., 2023; De et al., 2024), recurrent architectures with unitary transition matrices (Arjovsky et al., 2016; Jing et al., 2017), and optimizers such as Muon (Jordan et al., 2024; Liu et al., 2025). Across these settings, the common benefit is that unitary transformations preserve norms and make deep signal propagation easier to control.

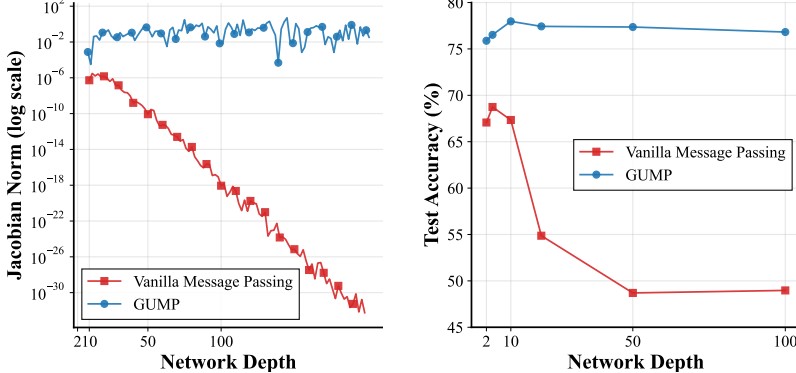

Figure 1: Compared to vanilla message passing (graph convolution), GUMP exhibits a stable Jacobian norm (left), which leads to more stable model performance as the number of layers increases (right).

However, constraining only the learnable parameters is insufficient for graph neural networks (GNNs), because repeated message passing is also governed by the graph structure itself. Nodes update their representations by aggregating information from neighbors (Gilmer et al., 2017), so the propagation dynamics depend on the adjacency operator rather than only on trainable weights. This creates graph-dependent forms of instability, most notably oversquashing and oversmoothing (Alon & Yahav, 2020; Topping et al., 2022; Chen et al., 2020a), which limit the benefits of depth for long-range reasoning on graphs.

Here, we extend the principle of unitarity from model parameters to the graph propagation operator itself. We propose Graph Unitary Message Passing (GUMP), a message-passing framework that replaces the usual normalized adjacency propagation with a unitary propagation operator on a transformed graph. The key point is not that all components of a GNN become unitary, but that the graph-induced factor in repeated propagation no longer decays exponentially with depth. This targets the graph-structure side of training instability, complementing standard practices that keep learnable weight matrices well conditioned.

To implement this idea, we transform a general graph into an Eulerian line-graph construction that admits unitary adjacency matrices while preserving the admissible edge-to-edge transitions induced by the original graph. We then use Newton-Schulz iteration (Kovarik, 1970; Björck & Bowie, 1971) to compute the unitary propagation matrix efficiently. Experiments across diverse graph learning tasks show that GUMP improves over vanilla message passing and strong baselines.

**Notations** We use bold uppercase letters for matrices, bold lowercase letters for vectors, and lowercase letters for scalars; for a matrix $\mathbf{X}$, $\mathbf{x}_i$ denotes row $i$ and $\mathbf{X}_{ij}$ an entry. The transpose and conjugate transpose are $\mathbf{X}^\top$ and $\mathbf{X}^\dagger$. Let $G = (V, E, \mathbf{X})$ be a graph with $n$ nodes, $e$ edges, and node features $\mathbf{X} \in \mathbb{R}^{n \times d}$; $\mathsf{V}[G]$ and $\mathsf{E}[G]$ denote its node and edge sets. We write $\tilde{\mathbf{A}}[G]$ for the binary adjacency, $\hat{\mathbf{A}}[G] = \mathbf{D}^{-1/2}\tilde{\mathbf{A}}[G]\mathbf{D}^{-1/2}$ for the normalized adjacency, and $\mathbf{A}[G]$ for a general matrix supported on $E$; when $G$ is clear, we omit the argument. The layer-$k$ GNN representation is $\mathbf{H}^{(k)} \in \mathbb{R}^{n \times d}$, with node vector $\mathbf{h}_i^{(k)}$. Additional graph-theory preliminaries are provided in the appendix.

## 2 Preliminaries on Unitarity and Training Instability

**Unitarity in Deep Learning** A matrix $\mathbf{U} \in \mathbb{C}^{n \times n}$ is *unitary* if $\mathbf{U}^\dagger\mathbf{U} = \mathbf{U}\mathbf{U}^\dagger = \mathbf{I}$; for real matrices this is orthogonality. Its key property is norm preservation, $\|\mathbf{U}\mathbf{x}\|_2 = \|\mathbf{x}\|_2$. This principle appears in orthogonal or unitary initialization (Saxe et al., 2013; Orvieto et al., 2023), unitary recurrent constraints for long-range sequence modeling (Arjovsky et al., 2016; Jing et al., 2017; Orvieto et al., 2023; De et al., 2024), and optimizers such as Muon (Jordan et al., 2024; Liu et al., 2025). In these settings, controlling the spectrum or norm of the transformation helps stabilize deep signal propagation.

**Training Instability in GNNs** To understand the training instability in GNNs, we first introduce the instability in RNNs from Figure 2(a). We simply formulate RNN as $\mathbf{h}_k = \sigma(\mathbf{W}\mathbf{h}_{k-1} + \mathbf{u}_k)$ with $\mathbf{h}_k$ being the hidden state at layer $k$, $\mathbf{W}$ being the transformation matrix, $\mathbf{u}_k$ being the input at time step $k$, and $\sigma$ being the activation function. The Jacobian of the hidden state with respect to the input is used to measure the training instability of RNN (Arjovsky et al., 2016). If the Jacobian norm $\|\partial\mathbf{h}_k/\partial\mathbf{u}_0\|_2$ grows exponentially with $k$, the gradient will explode, and if it decays exponentially, the gradient will vanish. When imposing unitarity on $\mathbf{W}$, the Jacobian norm is bounded, thus improves the training stability of RNNs.

For graphs, the training instability is mainly caused by the message passing mechanism. In Figure 2(a), a one-hop MPNN layer for GNN is formulated as $\mathbf{H}^{(k)} = \sigma(\mathbf{A}\mathbf{H}^{(k-1)}\mathbf{W}_k)$ for simplicity. Similar to RNN (Pascanu et al., 2013), the Jacobian $\partial\mathbf{H}^{(r)}/\partial\mathbf{x}$ (Topping et al., 2022) can measure the training instability of GNN. When the Jacobian is large, the gradient will explode, and when it is small, the gradient will vanish. Therefore, we are motivated to impose unitarity on $\mathbf{A}$ in GNN to improve learning efficiency.

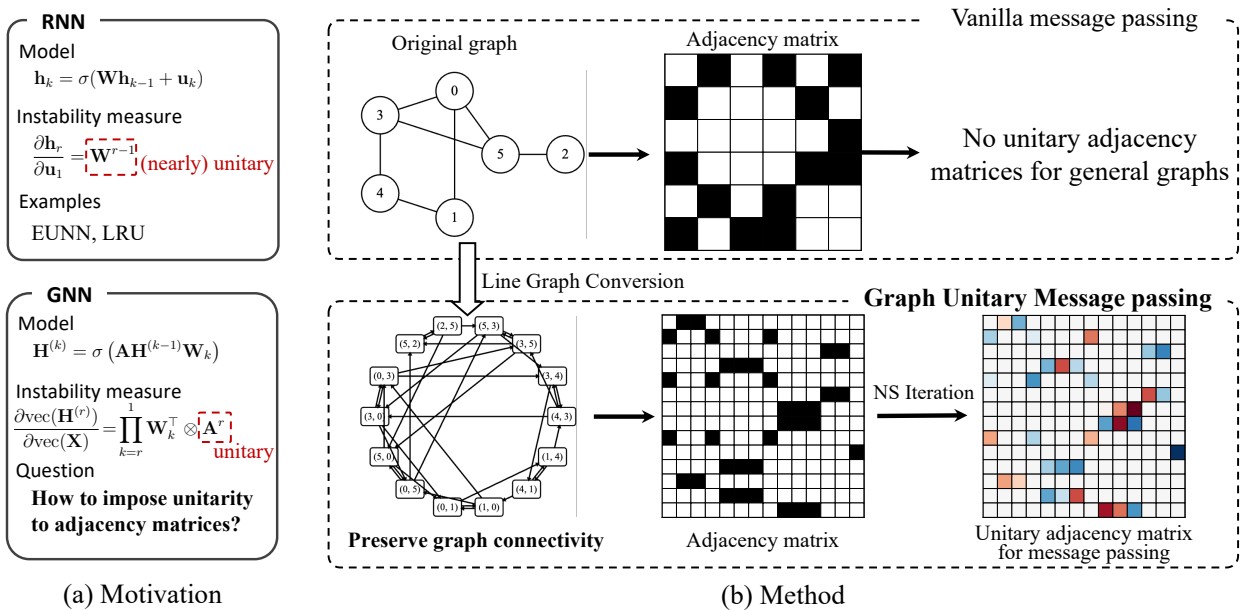

(a) Motivation            (b) Method

Figure 2: Overview of GUMP. (a) RNN versus GNN. The measures of the stability are derived with the identity activation function and $\otimes$ denotes the Kronecker product. (b) GUMP aims to impose unitarity on $\mathbf{A}$ in message passing. GUMP achieves this goal with graph transformation (Section 3.2) and unitary adjacency matrix calculation (Section 3.3).

# 3 Graph Unitary Message Passing

## 3.1 Challenges of Imposing Unitarity

Imposing unitarity on an adjacency matrix is not as straightforward as that in RNNs. The challenge comes from the sparsity of the adjacency matrix, as almost all unitary matrices are dense (Figure 2(b)). Therefore, the number of graphs with unitary adjacency matrices is limited. Also, the unitary adjacency matrix should be input-dependent and permutation-equivariant because the unitary adjacency matrix depends on input graph, and the order of nodes in a graph should not impact GNN representations.

In this section, the challenge of existence of unitary adjacency matrix is addressed by transforming the original graph to a special graph (Algorithm 1) which is guaranteed to have unitary adjacency matrices while preserving the admissible edge-to-edge transitions induced by the original graph at the same time. The challenge of preserving adjacency matrix property is addressed by calculating the unitary adjacency matrix with a unitary projection algorithm (Algorithm 2), which is implemented by utilizing the Newton-Schulz iteration and allows GUMP to be permutation-equivariant (Proposition 2). As a general one-hop message-passing mechanism, any convolution operation can be combined with GUMP by setting the edge weights to be the entries of the unitary adjacency matrix for message passing. The overview of GUMP is shown in Figure 2(b).

---

**Algorithm 1** Graph transformation

**Require:** A undirected graph $G = (V, E)$;
    Initialize a new digraph $G' = (V, E')$;
    **for** $(i, j) \in E$ **do**
       Add $(i, j)$ and $(j, i)$ to $E'$;
    **end for**
    Convert $G'$ to its line graph $\mathsf{L}(G')$;
    **Return:** A digraph $\mathsf{L}(G')$.

---

### 3.2 Graph Transformation: Convert Graph to Have Unitary Adjacency Matrix

Since unitary matrices are generally non-symmetric, the graph transformation algorithm should convert the original graph to a directed graph. We first formally define the unitary adjacency matrix as in Severini (2003). Given an adjacency matrix $\mathbf{A}$, its support matrix $\mathsf{S}[\mathbf{A}] \in \mathbb{R}^{n \times n}$ is a binary matrix with entries equal to one if the corresponding entry of $\mathbf{A}$ is non-zero and equal to zero otherwise, i.e., $\mathsf{S}[\mathbf{A}]_{ij} = 1$, if $\mathbf{A}_{ij} \neq 0$ and $\mathsf{S}[\mathbf{A}]_{ij} = 0$, if $\mathbf{A}_{ij} = 0$. Then, the unitary adjacency matrix $\mathbf{U}_G$ of graph $G$ is a unitary matrix whose support is equal to the support of its adjacency matrix $\mathbf{A}$, i.e., $\mathsf{S}[\mathbf{U}_G] = \mathsf{S}[\mathbf{A}]$.

We propose the transformation in Algorithm 1 for undirected graph $G$ to make it have unitary adjacency matrices while preserving the admissible edge-to-edge transitions induced by the original graph. Algorithm 1 first transforms the undirected graph into a digraph $G'$ by splitting each undirected edge into two directed edges. For each connected component of $G$ that contains at least one edge, the corresponding component of $G'$ is connected and has equal in-degree and out-degree at every vertex, hence that component is Eulerian. In particular, when $G$ is connected, $G'$ is Eulerian. Then, it converts the resulting digraph to its line graph $\mathsf{L}(G')$ (see definition in appendix). Finally, $\mathsf{L}(G')$ has unitary adjacency matrices because of its specularity and strong quadrangularity properties (Severini, 2003), which is proved in the following proposition.

**Proposition 1.** *The line graph $\mathsf{L}(G')$ returned by Algorithm 1 has unitary adjacency matrices.*

Proposition 1 is proved in the appendix. The proof treats disconnected inputs componentwise and combines the resulting unitary blocks by a block-diagonal direct sum. Algorithm 1 does not literally preserve node-level adjacency, because the vertices of $\mathsf{L}(G')$ correspond to directed edges of $G$. Rather, it preserves the admissible edge-to-edge transitions induced by the original graph: two vertices of $\mathsf{L}(G')$ are adjacent only when the corresponding directed edges in $G'$ can be traversed consecutively through a shared endpoint. Thus, the construction does not introduce spurious transitions across disconnected components or unrelated parts of the original graph. Finally, Algorithm 1 takes as input graph $G$ with $n$ nodes and $e$ edges, and outputs a line graph $\mathsf{L}(G')$ with $2e$ nodes. Its edge set contains exactly $\sum_{v \in V} d(v)^2$ directed edges in the undirected-to-Eulerian construction, because each vertex $v$ contributes all admissible transitions from its $d(v)$ incoming directed edges to its $d(v)$ outgoing directed edges. Therefore, constructing $\mathsf{L}(G')$ takes $\mathcal{O}\big(\sum_{v \in V} d(v)^2\big)$ time, which is $\mathcal{O}(|E|\Delta)$ for maximum degree $\Delta$ and $\mathcal{O}(|E|^2)$ in the worst case.

### 3.3 Unitary Adjacency Matrix Calculation: Compute the Edge Weights for Message Passing

According to Proposition 1, the line graph $\mathsf{L}(G')$ has unitary adjacency matrices. In this section, we propose an algorithm to calculate a unitary adjacency matrix for GUMP, because the unitary adjacency matrix depends on the input graph and should be calculated for each graph.

#### 3.3.1 Permutation-equivariant Projection

Permutation equivariance of message passing is a key property for GNN to apply to graphs with varying node orders. To achieve this, the calculation of unitary adjacency matrix has to be permutation equivariant. GUMP consists of two steps: 1) calculate edge weights to form weighted adjacency matrix; 2) impose unitarity on weighted adjacency matrix.

Firstly, edge weight for $(i, j) \in \mathsf{E}[\mathsf{L}(G')]$ is calculated by

$$\alpha_{ij} = \mathsf{Tanh}\left(\mathbf{w}^\top \cdot \mathsf{LeakyReLU}(\mathbf{W}_s \mathbf{h}_i + \mathbf{W}_t \mathbf{h}_j)\right), \tag{1}$$

where $\mathbf{h}_i$ ($\mathbf{h}_j$) is the representations for node $i$ ($j$) in $\mathsf{L}(G')$, $\mathbf{W}_s, \mathbf{W}_t \in \mathbb{R}^{d' \times d}$ are transformation matrices for source and target nodes of an edge respectively, and $\mathbf{w} \in \mathbb{R}^{d'}$ is a learnable parameter. Then, the weighted adjacency matrix of $\mathsf{L}(G')$, denoted as $\mathbf{A}_w \in \mathbb{R}^{2e \times 2e}$, is formed from edge weights, i.e., $(\mathbf{A}_w)_{ij} = \alpha_{ij}$.

After calculating $\mathbf{A}_w$, we impose unitarity on $\mathbf{A}_w$ by projection. We use the projection algorithm in Keller (1975), which takes advantage of the fact that the polar transformation yields the closest unitary matrix to a given matrix in terms of the Frobenius norm, i.e., $\mathsf{U}[\mathbf{A}_w] = \arg\min_{\mathbf{U} \text{ is unitary}} \|\mathbf{A}_w - \mathbf{U}\|_F^2 = \mathbf{A}_w \left(\mathbf{A}_w^\dagger \mathbf{A}_w\right)^{-\frac{1}{2}}$. The unitary projection $\mathsf{U}[\mathbf{A}_w]$ is guaranteed to be permutation-equivariant when $\mathbf{A}_w$ is a full-rank matrix with the following proposition.

**Proposition 2** (Strong permutation equivariance). *Given two permutation matrices* $\mathbf{P}_1$ *and* $\mathbf{P}_2$, *if* $\mathbf{A}_w$ *is a full-rank matrix, the unitary projection* $\mathsf{U}[\mathbf{A}_w]$ *is equivariant to both row and column permutations of* $\mathbf{A}_w$, *i.e.,* $\mathbf{P}_1\mathsf{U}[\mathbf{A}_w]\mathbf{P}_2^\top = \mathsf{U}[\mathbf{P}_1\mathbf{A}_w\mathbf{P}_2^\top]$.

By Proposition 2 (proved in appendix), the weighted adjacency matrix $\mathbf{A}_w$ should be full-rank to guarantee permutation equivariance of GUMP. Because the line-graph weighted adjacency decomposes into square transition blocks $\mathbf{B}_v$ as described below, this full-rank condition is equivalent to requiring every non-isolated block $\mathbf{B}_v$ to be full rank. Appendix D.1 reports reviewer-requested diagnostics on the five TU datasets: using tolerance $10^{-7}$, all transition blocks are full rank on MUTAG, ENZYMES, NCI1, and NCI109, while PROTEINS has a 99.99% full-rank rate. These results support the generic full-rank condition in the evaluated setting, but the formal equivariance statement still retains the full-rank assumption. However, the unitary projection is computationally expensive due to the inverse square root of $\mathbf{A}_w^\dagger\mathbf{A}_w \in \mathbb{R}^{2e \times 2e}$.

Before discussing the efficient implementation, we first clarify why the projection step is compatible with the sparse message-passing structure of $\mathsf{L}(G')$. Although polar factors are dense in general, the Eulerian line-graph construction gives $\mathbf{A}_w$ a special row/column block structure indexed by the intermediate vertex shared by two consecutive directed edges. The next proposition makes this structure explicit and shows that the projection preserves it.

**Proposition 3** (Admissible-transition support preservation). *Let* $\mathbf{A}_\mathsf{L}$ *and* $\mathbf{A}_w$ *be the binary and weighted adjacency matrices of* $\mathsf{L}(G')$, *where* $G' = (V, E')$ *is returned by Algorithm 1. If rows are grouped by incoming directed edges* $E_{\mathrm{in}}(v)$ *and columns by outgoing directed edges* $E_{\mathrm{out}}(v)$, *then there exist permutation matrices* $\mathbf{P}_{\mathrm{in}}$ *and* $\mathbf{P}_{\mathrm{out}}$ *such that*

$$\mathbf{D} := \mathbf{P}_{\mathrm{in}}\mathbf{A}_w\mathbf{P}_{\mathrm{out}}^\top = \mathsf{diag}(\mathbf{B}_v)_{v \in V},$$

*where* $\mathbf{B}_v$ *contains the weights for transitions* $E_{\mathrm{in}}(v) \to E_{\mathrm{out}}(v)$ *and is square because* $|E_{\mathrm{in}}(v)| = |E_{\mathrm{out}}(v)|$. *Under the same row and column permutations, the binary line-graph adjacency satisfies*

$$\mathbf{P}_{\mathrm{in}}\mathbf{A}_\mathsf{L}\mathbf{P}_{\mathrm{out}}^\top = \mathsf{diag}(\mathbf{J}_v)_{v \in V}, \qquad \mathbf{J}_v = \mathbf{1}_{d(v) \times d(v)},$$

*where zero-size blocks for isolated vertices are omitted and each dense all-ones block* $\mathbf{J}_v$ *represents all admissible transitions* $E_{\mathrm{in}}(v) \to E_{\mathrm{out}}(v)$. *If* $\mathbf{A}_w$ *is full-rank, then*

$$\mathbf{P}_{\mathrm{in}}\mathsf{U}[\mathbf{A}_w]\mathbf{P}_{\mathrm{out}}^\top = \mathsf{diag}(\mathsf{U}[\mathbf{B}_v])_{v \in V}.$$

*Since each projected block* $\mathsf{U}[\mathbf{B}_v]$ *lies inside the corresponding dense support block* $\mathbf{J}_v$, *every nonzero entry of* $\mathsf{U}[\mathbf{A}_w]$ *lies on an admissible transition of* $\mathsf{L}(G')$. *Hence* $\mathsf{S}[\mathsf{U}[\mathbf{A}_w]] \subseteq \mathsf{S}[\mathbf{A}_\mathsf{L}]$ *with equality whenever each block polar factor* $\mathsf{U}[\mathbf{B}_v]$ *is fully supported, which holds generically for continuous full-rank learned weights.*

Proposition 3 shows that GUMP does not rely on masking a dense polar factor after projection: the Eulerian line-graph construction induces the required block structure, and the polar projection preserves it.

### 3.3.2 Efficient Implementation of Unitarity Projection

Calculating the unitary projection $\mathsf{U}[\mathbf{A}_w] = \mathbf{A}_w(\mathbf{A}_w^\dagger\mathbf{A}_w)^{-\frac{1}{2}}$ directly involves computing the inverse square root of a matrix, which is computationally expensive and numerically unstable for large matrices, typically requiring singular value decomposition (SVD) with cubic complexity $\mathcal{O}((2e)^3)$. To address this, we employ Newton-Schulz iteration (Kovarik, 1970; Björck & Bowie, 1971; Jordan et al., 2024), an iterative method that converges quadratically to the unitary polar factor without expensive decompositions.

The Newton-Schulz iteration (Jordan et al., 2024) for computing the unitary factor $\mathbf{U}$ of a matrix $\mathbf{A}_w$ is given by the recurrence:

$$\mathbf{X}_{t+1} = \frac{15}{8}\mathbf{X}_t - \frac{5}{4}\mathbf{X}_t\left(\mathbf{X}_t^\top\mathbf{X}_t\right) + \frac{3}{8}\mathbf{X}_t\left(\mathbf{X}_t^\top\mathbf{X}_t\right)^2, \quad \mathbf{X}_0 = \frac{\mathbf{A}_w}{\|\mathbf{A}_w\|_F}. \tag{2}$$

Here, $\mathbf{X}_t$ converges to $\mathsf{U}[\mathbf{A}_w]$ as $t \to \infty$. The normalization by the Frobenius norm $\|\mathbf{A}_w\|_F$, which upper bounds the spectral norm, ensures the spectral norm is at most 1 for better convergence. Algorithm 2

uses a finite number of iterations and therefore returns an approximately unitary operator. Appendix D.1 reports the normalized residual $\|\mathbf{X}_{v,K}^{\top}\mathbf{X}_{v,K} - I\|_F/\sqrt{d_v}$ for $K = 5$ and $K = 10$. Increasing to $K = 10$ drives the median residual to around $10^{-7}$ on all five TU datasets, while the larger p95/max values are confined to the tail of ill-conditioned blocks. This is the expected accuracy–cost trade-off of a fixed-iteration polar solver; because the computation is blockwise, the effect is localized and can be controlled by increasing $K$ or applying a more accurate block projection only to flagged near-singular blocks. Because $\mathbf{A}_w$ can be written as a block diagonal matrix after the row/column permutations in Proposition 3, the Newton-Schulz iteration can be applied efficiently either by sparse matrix multiplications or by applying the iteration separately to each block $\mathbf{B}_v$ of $\mathbf{D}$ after the corresponding row/column permutations, making it significantly more efficient than SVD-based approaches for the sparse but large adjacency matrices encountered in GNNs. This yields an explicit complexity bound for the actual algorithm. Since Algorithm 1 replaces each undirected edge by both orientations, each block $\mathbf{B}_v$ has size $d(v) \times d(v)$, so every Newton–Schulz iterate decomposes into independent block updates. One iteration costs $\mathcal{O}\big(\sum_{v \in V} d(v)^3\big)$, and Algorithm 2 costs $\mathcal{O}\big(K \sum_{v \in V} d(v)^3\big)$ over $K$ iterations rather than $\mathcal{O}(K(2e)^3)$ for a naive dense implementation.

**Complexity and memory.** Let $M_L = \sum_{v \in V} d(v)^2$ be the number of admissible directed transitions in the line graph, $F$ the line-graph hidden dimension, and $F_a = d'$ the attention dimension in equation 1. The graph transformation costs $\mathcal{O}(e + M_L)$ time and storage per input graph; learned edge weights cost $\mathcal{O}(2eFF_a + M_LF_a)$ time and $\mathcal{O}(M_L)$ storage after source/target projections are precomputed. The $K$-step blockwise Newton–Schulz projection costs $\mathcal{O}\big(K \sum_{v \in V} d(v)^3\big)$ time and $\mathcal{O}(M_L)$ forward storage, although automatic differentiation may retain $\mathcal{O}(KM_L)$ intermediates without checkpointing or a custom backward pass. A GCN-style GUMP propagation layer then costs $\mathcal{O}(M_LF + 2eF^2)$, compared with $\mathcal{O}(eF + nF^2)$ for a standard GCN layer. Thus GUMP remains linear for bounded-degree sparse graphs but is degree-sensitive; high-degree vertices dominate $\sum_v d(v)^2$ and $\sum_v d(v)^3$.

**Comparison with Kiani et al.** Kiani et al. (2024) construct unitary graph convolutions on the original graph operator, for example through matrix-exponential unitary filters such as $\exp(i\mathbf{A})$. Unlike GUMP, they do not materialize a $2e$-node line graph or the $M_L = \sum_v d(v)^2$ edge-to-edge transition set, but their cost depends on how the unitary graph filters are constructed or approximated. Therefore, Kiani et al. avoids GUMP's line-graph expansion bottleneck, while its runtime depends on the construction or approximation of matrix-exponential unitary filters; a direct runtime comparison is implementation- and setup-dependent.

---

**Algorithm 2** Blockwise calculation of unitary adjacency matrix via Newton-Schulz iteration

---

**Require:** $\mathsf{L}(G')$ outputted by Algorithm 1, iterations $K$;
1: Calculate $\mathbf{A}_w$ of $\mathsf{L}(G')$ with equation 1;
2: Find $\mathbf{P}_{\text{in}}, \mathbf{P}_{\text{out}}$ such that $\mathbf{D} = \mathbf{P}_{\text{in}}\mathbf{A}_w\mathbf{P}_{\text{out}}^{\top} = \mathsf{diag}(\mathbf{B}_v)_{v \in V}$ as in Proposition 3;
3: **for** each block $\mathbf{B}_v$ **do**
4:     Normalize $\mathbf{X}_{v,0} = \mathbf{B}_v/\|\mathbf{B}_v\|_F$;
5:     **for** $t = 0$ to $K - 1$ **do**
6:         $\mathbf{X}_{v,t+1} = \frac{15}{8}\mathbf{X}_{v,t} - \frac{5}{4}\mathbf{X}_{v,t}\big(\mathbf{X}_{v,t}^{\top}\mathbf{X}_{v,t}\big) + \frac{3}{8}\mathbf{X}_{v,t}\big(\mathbf{X}_{v,t}^{\top}\mathbf{X}_{v,t}\big)^2$;
7:     **end for**
8: **end for**
9: $\mathsf{U}[\mathbf{D}] = \mathsf{diag}(\mathbf{X}_{v,K})_{v \in V}$;
10: $\mathsf{U}[\mathbf{A}_w] = \mathbf{P}_{\text{in}}^{\top}\mathsf{U}[\mathbf{D}]\mathbf{P}_{\text{out}}$;
11: **Return:** Approximately unitary adjacency matrix $\mathsf{U}[\mathbf{A}_w]$.

---

## 3.4 Applying GUMP to GNNs

**Architecture and scatter operation.** We can apply GUMP to different GNN architectures for graph learning tasks in Algorithm 3. Given a graph $G$, a base GNN first computes the initial node representations of $G$, i.e., $\mathbf{X}^{(0)} = \mathsf{GNN}(\mathbf{X}, G)$. Then, Algorithm 1 transforms $G$ to $\mathsf{L}(G')$. The initial node representations $\mathbf{H}^{(0)} \in \mathbb{R}^{2e \times 2d}$ of $\mathsf{L}(G')$ are generated with $\mathbf{h}_{(i,j)} = [\mathbf{x}_i^{(0)}; \mathbf{x}_j^{(0)}]$, $\forall(i,j) \in \mathsf{V}[\mathsf{L}(G')]$ $(i, j \in \mathsf{V}[G])$. Next, the unitary adjacency matrix $\mathsf{U}[\mathbf{A}_w]$ of $\mathsf{L}(G')$ is calculated from Algorithm 2 and applied to propagate messages

---

**Algorithm 3** GNN with the graph unitary message passing mechanism (GNN-GUMP)

---

**Require:** A graph $G = (V, E, \mathbf{X})$;

1: $\mathbf{X}^{(0)} = \mathsf{GNN}(\mathbf{X}, G)$;

2: Transform $G$ to $\mathsf{L}(G')$ with Algorithm 1;

3: Generate initial representation $\mathbf{H}^{(0)}$ for $\mathsf{L}(G')$ with $\mathbf{h}_{(i,j)} = [\mathbf{x}_i^{(0)}; \mathbf{x}_j^{(0)}], \forall (i,j) \in \mathsf{V}[\mathsf{L}(G')]$;

4: Calculate $\mathsf{U}[\mathbf{A}_w]$ with Algorithm 2;

5: **for** $k = 1 \cdots L$ **do**

6: $\quad \mathbf{h}_v^{(k)} = \gamma(\mathbf{h}_v^{(k-1)}, \sum_{u \in \mathcal{N}_{\mathsf{L}(G')}(v)} \mathsf{U}[\mathbf{A}_w]_{vu} \phi^{(k)}(\mathbf{h}_v^{(k-1)}, \mathbf{h}_u^{(k-1)})), \, v \in \mathsf{V}[\mathsf{L}(G')]$

7: **end for**

8: Scatter $\mathbf{H}^{(L)}$ to nodes of $G$ with $\mathbf{H}_s^{(L)} = \mathsf{Scatter}(\mathbf{H}^{(L)}, G)$;

9: Generate node representations of $G$ with $\mathbf{X}^{(L)} = [\mathbf{X}^{(0)}; \mathbf{H}_s^{(L)}]$.

10: **Return:** Node representations $\mathbf{X}^{(L)}$ of $G$.

---

with $\mathbf{h}_v^{(k)} = \gamma(\mathbf{h}_v^{(k-1)}, \sum_{u \in \mathcal{N}_{\mathsf{L}(G')}(v)} \mathsf{U}[\mathbf{A}_w]_{vu} \phi^{(k)}(\mathbf{h}_v^{(k-1)}, \mathbf{h}_u^{(k-1)})), \, v \in \mathsf{V}[\mathsf{L}(G')]$, where $\mathcal{N}_{\mathsf{L}(G')}(v)$ denotes the line-graph neighborhood of $v$. Here, $\phi^{(k)}(\cdot, \cdot)$ denotes a generic layer-$k$ message function, and $\gamma(\cdot, \cdot)$ denotes the generic combine/update function that merges a node's previous representation with the aggregated message. In the method itself, GUMP constrains only the propagation operator $\mathsf{U}[\mathbf{A}_w]$; the learnable matrices inside the base GNN remain unconstrained. In the theory, we separately assume that the cumulative contribution of the weight matrices remains well conditioned, so that the analysis isolates the effect of repeated graph propagation. After $L$ layers of unitary message passing, we obtain the node representations $\mathbf{H}^{(L)}$, which is later scattered to nodes of $G$ with $\mathbf{H}_s^{(L)} = \mathsf{Scatter}(\mathbf{H}^{(L)}, G) \in \mathbb{R}^{n \times d'}$. Then, $\mathbf{H}_s^{(L)}$ are concatenated with $\mathbf{X}^{(0)}$ to obtain the final node representations $\mathbf{X}^{(L)} = [\mathbf{X}^{(0)}; \mathbf{H}_s^{(L)}] \in \mathbb{R}^{n \times (d+d')}$ of $G$. Finally, various graph learning tasks, e.g., graph and node classification, link prediction, and graph regression, are performed based on $\mathbf{X}^{(L)}$. In this paper, GUMP is a general one-hop message-passing mechanism for GNN. Therefore, depending on the convolution in line 6 of Algorithm 3, GNN with GUMP is named as [GNN type]-GUMP in Section 4, e.g., GCN-GUMP and GIN-GUMP have graph convolution and isomorphism operators in line 6 of Algorithm 3, respectively.

Each node of $\mathsf{L}(G')$ corresponds to a directed edge $(u, v)$ in the bidirected graph $G'$. We map directed-edge representations back to original nodes by aggregating the line-graph states whose directed edges terminate at the node. In our implementation, this scatter operation is a mean aggregation:

$$\mathbf{h}_{s,v}^{(L)} = \frac{1}{|\mathcal{I}_{\text{in}}(v)|} \sum_{(u,v) \in \mathcal{I}_{\text{in}}(v)} \mathbf{h}_{(u,v)}^{(L)},$$

where $\mathcal{I}_{\text{in}}(v)$ denotes the set of incoming directed edges to $v$. For isolated nodes, which create no line-graph vertices, the scatter output is set to zero. We concatenate $\mathbf{H}_s^{(L)}$ with $\mathbf{X}^{(0)}$ as a residual node-level path. This preserves original node information that may not be represented by the edge-centric line graph, especially for isolated nodes that create no line-graph vertices. The concatenation also keeps local node-level features available to the task head while $\mathbf{H}_s^{(L)}$ contributes long-range unitary edge-to-edge propagation features.

**Expected-Jacobian motivation and row-energy stability.** To connect the propagation analysis to training stability, consider the simplified message-passing dynamics

$$\mathbf{H}^{(\ell)} = \mathsf{ReLU}(\mathbf{A}\mathbf{H}^{(\ell-1)}\mathbf{W}_\ell), \qquad \mathbf{H}^{(0)} = \mathbf{X}.$$

The following factorization separates the cumulative feature-transformation term from the graph-propagation term.

**Proposition 4** (Expected-Jacobian factorization)**.** *Under the path-level random-gate approximation with a non-graph gate factor $\rho_L$, the expected Jacobian from source node $s$ to target node $i$ satisfies*

$$\mathbb{E}\left[\frac{\partial \mathbf{h}_i^{(L)}}{\partial \mathbf{x}_s}\right] = \rho_L \left(\prod_{\ell=L}^{1} \mathbf{W}_\ell^\top\right) (\mathbf{A}^L)_{is}.$$

Here $\rho_L$ is an analysis-only path-level factor summarizing ReLU gate effects along length-$L$ computation paths. It is treated as independent of the graph propagation operator. Proposition 4 shows that, after separating the cumulative weight term and the non-graph gate factor, the graph-dependent depth behavior of the expected Jacobian is captured by the scalar propagation factor $(\mathbf{A}^L)_{is}$.

For a fixed target node $i$, we therefore measure the source-averaged squared graph factor by

$$I_i(L; \mathbf{A}) := \frac{1}{N} \sum_{s=1}^{N} |(\mathbf{A}^L)_{is}|^2 = \frac{1}{N} \|\mathbf{e}_i^\top \mathbf{A}^L\|_2^2.$$

Equivalently,

$$\frac{1}{N} \sum_{s=1}^{N} \left\| \mathbb{E}\left[ \frac{\partial \mathbf{h}_i^{(L)}}{\partial \mathbf{x}_s} \right] \right\|_F^2 = |\rho_L|^2 I_i(L; \mathbf{A}) \left\| \prod_{\ell=L}^{1} \mathbf{W}_\ell^\top \right\|_F^2.$$

Thus, $I_i(L; \mathbf{A})$ is not an additional architectural objective; it is the source-averaged graph factor appearing in the expected Jacobian.

Classical oversmoothing analyses often study the decay of non-stationary spectral components under repeated normalized propagation. In our setting, Proposition 4 shows that the graph-dependent factor of the expected Jacobian is exactly $(\mathbf{A}^L)_{is}$. We therefore study its source-averaged squared magnitude, $I_i(L; \mathbf{A})$, which removes sign/phase cancellations and measures the row energy of the graph-propagation contribution. Theorem 1 shows that unitary propagation preserves this aggregate factor, while Theorem 2 recovers the usual exponential decay of non-stationary modes for vanilla normalized propagation.

**Theorem 1** (Row-energy stability of unitary graph propagation). *Let $\mathbf{A} \in \mathbb{C}^{N \times N}$ be unitary. For every target node $i \in [N]$ and depth $L \geq 0$,*

$$I_i(L; \mathbf{A}) = \frac{1}{N} \|\mathbf{e}_i^\top \mathbf{A}^L\|_2^2 = \frac{1}{N}.$$

*Equivalently, unitary propagation preserves the source-averaged graph factor in the expected Jacobian. Hence, under the random-gate approximation and the well-conditioned-weight assumption, the graph propagation operator does not introduce exponential decay or explosion in this aggregate Jacobian measure.*

Theorem 1 should be interpreted as a row-energy conservation result. Since $\mathbf{A}$ is unitary, $\mathbf{A}^L$ is also unitary for every depth $L$, so the squared $\ell_2$ norm of each row of $\mathbf{A}^L$ is preserved. Under the standard assumption that the cumulative weight factor $\prod_{\ell=L}^{1} \mathbf{W}_\ell^\top$ remains well conditioned and does not itself scale exponentially with $L$, any exponential decay or explosion in the expected Jacobian cannot come from the graph-propagation term. This is the intended claim of Theorem 1.

The theorem does not claim that every individual entry $(\mathbf{A}^L)_{is}$ is constant, uniformly lower bounded, or that the full Jacobian is depth-invariant. Individual source-target entries may increase, decrease, or oscillate because of sign or phase cancellations. The invariant is the row-energy aggregate $I_i(L; \mathbf{A})$, which is the source-averaged graph factor in the expected Jacobian.

**Theorem 2** (Vanilla spectral decay). *For vanilla message passing with $\mathbf{A} = \hat{\mathbf{A}} \in \mathbb{R}^{N \times N}$, assume $\hat{\mathbf{A}}$ is real symmetric with spectral radius 1 and eigenvalues*

$$1 = \lambda_1 > |\lambda_2| \geq \cdots \geq |\lambda_N|, \qquad c := \max_{j \geq 2} |\lambda_j| < 1.$$

*Let $\{v_j\}_{j=1}^{N}$ be an orthonormal eigenbasis and $\mathbf{\Pi} := v_1 v_1^\top$. Then for every target node $i \in [N]$ and depth $L \geq 0$,*

$$I_i(L; \hat{\mathbf{A}}) = \mathbb{E}_{s \sim \mathrm{Unif}([N])} \left[ \left| (\hat{\mathbf{A}}^L)_{is} \right|^2 \right] = \frac{1}{N} v_{1,i}^2 + \frac{1}{N} \sum_{j=2}^{N} v_{j,i}^2 \lambda_j^{2L},$$

*and therefore*

$$0 \leq I_i(L; \hat{\mathbf{A}}) - \frac{1}{N} v_{1,i}^2 \leq \frac{1}{N} c^{2L}.$$

*Equivalently, if $\delta_L(i,s) := (\hat{\mathbf{A}}^L - \mathbf{\Pi})_{is}$, then*

$$\mathbb{E}_s\left[|\delta_L(i,s)|^2\right] = \frac{1}{N}\sum_{j=2}^{N} v_{j,i}^2 \lambda_j^{2L} \leq \frac{1}{N}c^{2L}.$$

*Moreover, with $\alpha_i := \sum_{j:\,|\lambda_j|=c} v_{j,i}^2$, we also have $\mathbb{E}_s\left[|\delta_L(i,s)|^2\right] \geq \frac{1}{N}\alpha_i c^{2L}$. Thus, if $\alpha_i > 0$, the non-stationary source-averaged influence decays as $\Theta(c^{2L}/N)$.*

Theorem 2 shows that vanilla normalized propagation preserves only the stationary component, while the non-stationary source-averaged graph influence decays exponentially with depth. In contrast, Theorem 1 shows that unitary propagation preserves $I_i(L;\mathbf{A}) = 1/N$, so GUMP removes the graph-induced exponential decay in this aggregate Jacobian factor. Proof is in Appendix C.5.

## 4 Experiments

In this section, we perform experiments to evaluate GUMP on graph learning tasks. All experiments are implemented by PyTorch Geometric (Fey & Lenssen, 2019) and conducted on NVIDIA RTX 4090 GPUs and AMD EPYC 7763 CPUs.

### 4.1 Experiments on Synthetic Dataset

**Setup** In this section, we conduct experiments on synthetic datasets, i.e., CrossedRing, Ring, and CliquePath, in Di Giovanni et al. (2023) to test GUMP. The performance is evaluated on the distances from source to target in the range of 4 to 28. This experiment is designed to test the ability of GUMP to learn long-range interactions when increasing the model layers. In the experiments, we compare GCN-GUMP and GCN. The layer $L$ of GCN-GUMP and GCN is appropriately set up according to the distance $d$ between source and target in the synthetic datasets (i.e., $L = \lfloor d/2 \rfloor + 1$), such that the long-range interactions can be captured by GNN. We set the hidden dimension to be 32 for both GCN-GUMP and GCN. The hyperparameters of GCN-GUMP for synthetic datasets are in appendix.

**Results** We plot the average results from three random seeds of GCN-GUMP and GCN experiments in Figure 3. For two easier datasets, i.e., CrossedRing and Ring, GUMP achieves 100% accuracy when the distance ranges from 4 to 28. For the challenging CliquePath dataset, GCN-GUMP's performance deteriorates to random guessing at a distance of 28. The results show that GUMP can help capture the long-range interactions in graph learning tasks. We compare with more baselines in appendix.

### 4.2 Experiments on the TUDataset

**Datasets** We select five TUDataset benchmarks (Morris et al., 2020): Mutag, Proteins, Enzymes, NCI1, and NCI109. These chemistry or biological graphs contain complex interactions, including cases where atoms far apart in graph distance may be close in space. Dataset statistics are in the appendix.

**Baselines** Baselines include various rewiring methods, i.e., DIGL (Gasteiger et al., 2019), SDRF (Topping et al., 2022), FoSR (Karhadkar et al., 2023), GTR (Black et al., 2023), and ELDANADD (Jamadandi et al., 2024), diffusion methods, i.e., ADGN (Tönshoff et al., 2023) and GRAND (Chamberlain et al., 2021), transformers method GPS (Rampášek et al., 2022), GNN with orthogonal parameters Ortho-GConv (Guo et al., 2022), k-hop message passing GRIT (Di Giovanni et al., 2023), and Kiani et al. (Kiani et al., 2024). The baselines' settings follow Karhadkar et al. (2023). We refer to the unitary-convolution baseline of Kiani et al. (2024) as "Kiani et al." in all tables for consistency. For the Kiani et al. entries on these five TU datasets, we reproduce the results by running the released code of Kiani et al. (2024) under the TU evaluation protocol.

**Experimental Details** For each method, 10% of graphs are held out for testing and the remaining 90% form the development set. We use 100 random train/validation splits of the development set, with 80% for

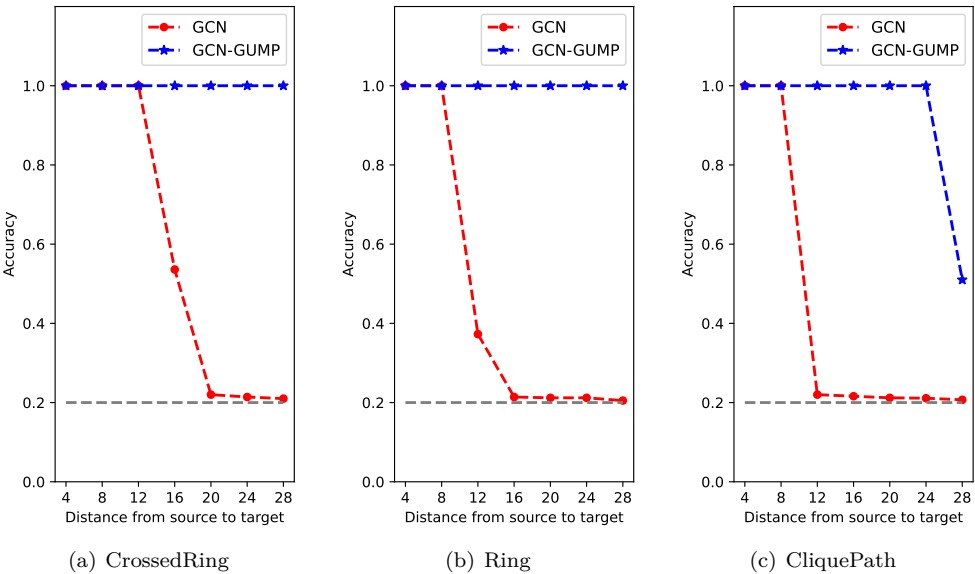

Figure 3: The performance of GCN and GCN-GUMP on the CrossedRing, Ring, and CliquePath with different distances from source to target.

Table 1: Graph classification accuracy on the TUDataset. **First** and second best results are bold and underlined, respectively.

| Classes | Methods | Mutag | Proteins | Enzymes | NCI1 | NCI109 |
|---------|---------|-------|----------|---------|------|--------|
| – | GIN
GIN (+layer) | 77.70±3.60
69.80±2.75 | 70.80±0.83
68.71±0.96 | 33.80±1.12
25.92±1.07 | 75.65±0.49
73.49±0.46 | 74.93±0.46
72.47±0.53 |
| Rewiring (GIN) | DIGL
SDRF
FoSR
GTR
ELDANADD | 79.80±2.08
78.40±2.80
78.00±2.22
77.60±2.84
82.16±0.03 | 70.71±0.67
69.81±0.79
75.11±0.82
73.13±0.69
70.53±0.86 | 35.74±1.20
35.82±1.09
29.20±1.38
30.57±1.42
26.36±0.01 | 79.37±0.43
74.55±0.54
70.15±0.47
75.45±0.44
– | 76.88±0.39
73.89±0.43
69.93±0.45
75.28±0.42
– |
| Diffusion | ADGN
GRAND | 81.39±1.81
77.94±1.73 | 73.81±0.80
73.24±0.94 | 28.78±1.25
24.13±1.05 | 76.15±0.42
68.51±0.48 | 74.31±0.44
67.26±0.46 |
| Transformer | GPS | 70.12±1.68 | 69.32±0.86 | 30.43±1.76 | 61.32±0.51 | 61.01±0.22 |
| – | Ortho-GConv | 71.78±2.52 | 63.80±0.98 | 18.30±1.13 | 69.92±0.60 | 68.91±0.50 |
| – | Kiani et al. | 78.00±1.96 | 71.57±0.60 | 43.67±1.45 | 78.71±0.42 | 78.10±0.34 |
| K-hop MP | GRIT | 80.76±2.18 | 73.71±0.89 | 35.22±1.17 | 72.21±0.46 | 71.68±0.44 |
| Ours | GIN-GUMP | **87.11±7.61** | **77.53±2.34** | **53.11±5.80** | **80.78±1.57** | **80.50±1.63** |

training and 10% for validation, stop with patience 100 based on validation loss, and report test accuracy with 95% confidence intervals for the best validation setting across runs. GUMP depth is manually tuned because capturing long-range interactions requires sufficient layers; detailed hyperparameters are in the appendix.

**Results** The results of GUMP on TUDataset are shown in Table 1. GUMP achieves superior performance across all five datasets, with GIN-GUMP obtaining the best results, demonstrating that the unitary message passing mechanism effectively learns complex interactions in graphs. Moreover, since GUMP usually has more layers than baselines in these datasets, experiments show that GCNs with more layers exhibit degraded performance on all datasets, showing that improvement of GUMP does not come from increasing expressivity with more GNN layers.

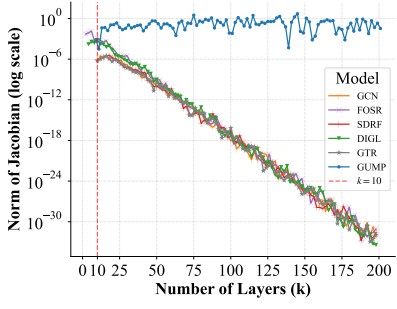

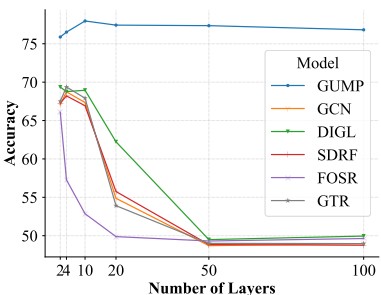

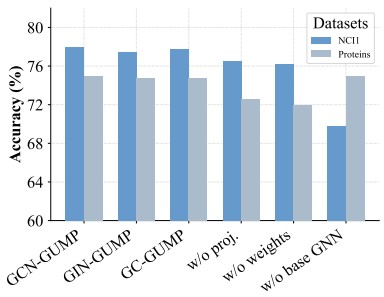

(a) Jacobian norm versus layers on NCI1     (b) Accuracy versus layers on NCI1     (c) Ablation on different components

Figure 4: Model analysis. GCN, GIN, and GUMP in (c) represent the convolution of GUMP. The base GNN of GUMP is GCN. "w/o proj" removes unitary projection in GUMP. "w/o weights" removes weighted adjacency matrix and unitary projection in GUMP. "w/o base GNN" removes base GNN in GUMP.

## 4.3 Experiments on LRGB

We evaluate on the Peptides-func and Peptides-struct datasets from the Long Range Graph Benchmark (LRGB) (Dwivedi et al., 2022); dataset statistics are in the appendix. Following Tönshoff et al. (2023), we compare GUMP with reproduced long-range graph baselines, including SDRF, FoSR, GTR, LASER (Barbero et al., 2023), GRAND, ADGN, GPS, Exphormer (Shirzad et al., 2023), GRIT, Graph ViT and G-MLPMixer (He et al., 2023), Drew-GCN (Gutteridge et al., 2023), ProxyAdd+GCN (Jamadandi et al., 2024), and k-GCN-SSM (Arroyo et al., 2025). For GUMP, the base one-hop operator is GCN; the appendix reports all tuned layers, optimization settings, and training hyper-

Table 2: Results of Peptides-func and Peptides-struct. **Bold** and underline denote best and second-best results.

| Methods | Peptides-func Test AP ↑ | Peptides-struct Test MAE ↓ |
|---|---|---|
| GCN | 0.6860±0.0050 | 0.2460±0.0007 |
| SDRF | 0.6874±0.0032 | 0.2453±0.0018 |
| FoSR | 0.6878±0.0030 | 0.2461±0.0016 |
| GTR | 0.6740±0.0033 | 0.2509±0.0013 |
| LASER | 0.6440±0.0010 | 0.3043±0.0019 |
| GRAND | 0.5789±0.0062 | 0.3418±0.0015 |
| ADGN | 0.5975±0.0044 | 0.2874±0.0021 |
| GPS | 0.6534±0.0091 | 0.2509±0.0014 |
| Exphormer | 0.6527±0.0043 | 0.2481±0.0007 |
| GRIT | 0.6988±0.0082 | 0.2460±0.0012 |
| Graph ViT | 0.6942±0.0075 | 0.2449±0.0016 |
| G-MLPMixer | 0.6921±0.0054 | 0.2475±0.0015 |
| Drew-GCN | 0.6996±0.0076 | 0.2781±0.0028 |
| ProxyAdd+GCN | 0.6789±0.0002 | 0.2465±0.0004 |
| k-GCN-SSM | 0.6902±0.0138 | 0.2898±0.0324 |
| Kiani et al. | 0.7072±0.0035 | **0.2425±0.0009** |
| GUMP | **0.7088±0.0026** | 0.2438±0.0014 |

parameters. We also include the Kiani et al. results reported by Kiani et al. (2024) as a recent unitary-convolution baseline, noting that those numbers follow that paper's setup rather than our reproduced protocol. The hyperparameters of GUMP and more results for LRGB are presented in appendix.

The results of LRGB are shown in Table 2. GUMP achieves the best result on Peptides-func and the second-best result on Peptides-struct, improving over GCN on both datasets while remaining slightly behind the reported Kiani et al. result on Peptides-struct. Overall, GUMP is competitive with recent unitary-convolution methods on LRGB while using a sparse, input-dependent unitary message-passing construction.

## 4.4 Model Analysis

We analyze GUMP through Jacobian norms, depth, and ablations.

**Jacobian** We visualize the spectral norm of the Jacobian for a distance-ten node pair from NCI1, using GCN as the base GNN. Figure 4(a) shows that GUMP remains stable as depth increases, while the baselines decay rapidly. Although this operator-norm diagnostic differs from the row-energy quantity $I_i(L; \mathbf{A})$, both probe repeated graph propagation: normalized propagation suppresses non-stationary modes, whereas uni-

tary propagation preserves their magnitudes. The trend is therefore directionally consistent with Theorems 1 and 2. DIGL has a larger Jacobian norm than other rewiring methods for depths below 50, matching its stronger performance in Table 1.

**Deep GNN** We increase the number of layers of GUMP and baselines to test depth robustness. This experiment is conducted in NCI1 with base GNN as GCN and the number of layers in the range of 2 to 100. The results in Figure 4(b) demonstrate that the performance of GUMP increases from 75.88% to 77.97% when increasing the number of layers from 2 to 10. However, baseline performance decreases substantially with depth. For example, the performance of FoSR decreases from 66.06% to 52.86% when the number of layers increases from 2 to 10. The baselines change drastically as layers increase, while GUMP is more stable, indicating that it can use deeper propagation for long-range interactions.

**Ablation Studies** Figure 4(c) analyzes GUMP components. Replacing the GUMP convolution with GIN or GraphConv (Morris et al., 2019) changes performance, showing sensitivity to the underlying convolution. Removing the unitary projection (message passing with $\mathbf{A}_w$) or both weights and projection (message passing with $\tilde{\mathbf{A}}[\mathsf{L}(G')]$) degrades performance, supporting their importance. Removing the base GNN decreases performance on NCI1 but not on Proteins, consistent with the dependence of $\mathbf{A}_w$ on line-graph node representations in equation 1.

Overall, GUMP shows the most depth-stable Jacobian behavior among the compared methods. Appendix D reports dataset statistics, hyperparameters, additional comparisons, and Jacobian visualizations; Appendix D.3 reports node-classification and runtime results, including Table 10, which motivates the scalability limitation in Section 5.

## 5 Discussions and Limitations

In this paper, we propose Graph Unitary Message Passing (GUMP), which propagates messages using unitary adjacency matrices and performs well across several graph learning tasks. We discuss below its limitations and future work.

**Scalability to large graphs.** The current implementation is less suitable for large single-graph settings and high-degree vertices. The line-graph transformation expands an input with $e$ undirected edges into $2e$ directed-edge states and $M_L = \sum_{v \in V} d(v)^2$ transitions; message passing stores $\mathcal{O}(2eF)$ states and processes $\mathcal{O}(M_L F)$ messages per layer, while blockwise projection costs $\mathcal{O}(K \sum_v d(v)^3)$. Appendix D.3, Table 10 shows that GUMP is slower than GCN under a matched propagated-node mini-batch budget, so GUMP is most appropriate when depth stability or long-range propagation justifies this extra cost, especially on small- and medium-scale graph classification or bounded-degree sparse graphs. Reducing the projection's dependence on high-degree vertices remains important for scaling GUMP. Scaling will likely require line-graph sampling, degree-aware transition sparsification, or approximate block-unitary propagation.

**Finite-step Newton–Schulz approximation.** Our stability analysis uses the exact polar factor $\mathsf{U}[\mathbf{A}_w]$, while Algorithm 2 returns an approximately unitary operator after $K$ Newton–Schulz iterations. The finite iterates preserve admissible block support and improve conditioning, but the residuals in Appendix D.1 show an accuracy–cost trade-off on ill-conditioned blocks. The larger residuals are concentrated in the ill-conditioned tail under a fixed iteration budget, so they reflect numerical accuracy rather than a structural failure of the construction. Because projection is blockwise, this numerical effect is local and can be controlled by increasing $K$ or selectively applying a higher-accuracy projection to near-singular blocks. A complete theory should quantify how residual non-unitarity accumulates with depth and depends on block condition numbers and graph degrees.

**Information Loss** For undirected, unweighted graphs, GUMP's transformation is lossless. For weighted or directed graphs, edge features encode weights or directions following R-GNNs (Battaglia et al., 2018), and post-processing filters invalid directed edges. These adaptations preserve the essential graph information used by GUMP while acknowledging that weighted and directed settings require explicit edge attributes and task-specific filtering.

## Acknowledgements

This work is supported by National Key Research and Development Program of China (under Grant No. 2023YFB2903904) and Beijing Science and Technology Program (No.Z251100008125003).

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

# A    Related work

## A.1    GNN and its Training Instability

Graph neural networks (GNNs) (Kipf & Welling, 2016; Gilmer et al., 2017) learn node representations through message passing, but deep propagation is often limited by oversquashing and oversmoothing. Oversquashing was highlighted by Alon & Yahav (2020), and Topping et al. (2022) connected it to Jacobian-based sensitivity and proposed curvature-inspired rewiring. Subsequent work has alleviated oversquashing by improving spectral gap or connectivity properties, including expander-style rewiring (Banerjee et al., 2022), FoSR (Karhadkar et al., 2023), DiffWire (Arnaiz-Rodríguez et al., 2022), total-resistance-based rewiring (Black et al., 2023), commute-time analysis (Di Giovanni et al., 2023), locality-preserving rewiring (Barbero et al., 2023), and multi-hop architectures such as Drew (Gutteridge et al., 2023).

Oversmoothing refers to the tendency of node representations to become indistinguishable as depth increases (Li et al., 2018; Oono & Suzuki, 2019). This effect has been studied empirically and theoretically (Kipf & Welling, 2016; Wu et al., 2020), with remedies including structure modification such as DropEdge and diffusion-based sparsification (Rong et al., 2019; Gasteiger et al., 2019), normalization methods such as PairNorm (Zhao & Akoglu, 2019), and architectural changes such as residual connections, Jumping Knowledge, and hierarchical pooling (Li et al., 2019a; Chen et al., 2020b; Xu et al., 2018; Gao & Ji, 2019). Recent work also studies oversquashing and oversmoothing jointly (Jamadandi et al., 2024; Arroyo et al., 2025).

## A.2    Line graphs and edge-centric GNNs.

Graph neural networks on line graphs and related edge-centric formulations have been studied to better align representation learning with edge-level tasks such as link prediction. In a line graph, each edge in the original graph is represented as a node, allowing message passing to operate directly over links rather than first learning endpoint embeddings and then applying a separate decoder. Early dual or edge-node co-embedding architectures, such as Dual-Primal Graph Convolutional Networks (Monti et al., 2018) and CensNet (Jiang et al., 2019), propagate information over both vertex- and edge-level structures, with CensNet explicitly using the line graph to switch the roles of nodes and edges. Edge-aware architectures such as NENN (Yang & Li, 2020) and EGAT (Wang et al., 2021) further model interactions between node and edge features, although they are not primarily designed as line-graph link-prediction methods. For link prediction, SEAL (Zhang & Chen, 2018) established the effectiveness of extracting enclosing subgraphs around candidate links and learning structural heuristics with a GNN, but it treats each candidate as a graph-classification instance. LGLP (Cai et al., 2022) instead directly transforms the enclosing subgraph of a candidate link into a line graph, reducing link prediction to node classification on the corresponding edge-node. LGCL (Zhang et al., 2023) extends this direction by introducing contrastive learning between the original subgraph view and the line-graph view, demonstrating that line-graph representations capture complementary edge-neighborhood information useful for predicting missing links. Recent preprint work has also explored directed link prediction by constructing directed line graphs and fusing local and global features (Zhang et al., 2025), suggesting a possible extension of line-graph GNNs beyond undirected settings. In parallel, strong link-prediction models such as WalkPooling (Pan et al., 2022) and ELPH/BUDDY (Chamberlain et al., 2023) learn subgraph-aware or edge-centric representations without explicitly materializing the full line graph, offering more scalable alternatives when line-graph construction becomes expensive around high-degree nodes. Overall, existing work suggests that line-graph GNNs are attractive for link prediction because they make edges the primary objects of message passing, but their practical use must balance task alignment against scalability, directionality, multigraph handling, and the cost of constructing dense edge-to-edge neighborhoods.

## A.3    Unitarity in Deep Learning

Unitarity has been widely used to stabilize deep models. In recurrent networks, unitary and orthogonal parameterizations such as uRNN, EUNN, scoRNN, expRNN, projUNN, and LRU improve long-range signal propagation by controlling the spectrum of the transition operator (Arjovsky et al., 2016; Jing et al., 2017; Helfrich et al., 2018; Lezcano-Casado & Martınez-Rubio, 2019; Kiani et al., 2022; Orvieto et al., 2023).

Related recurrent architectures, including RWKV and Mamba, share the broader goal of maintaining stable long-range dynamics (Peng et al., 2023; Gu & Dao, 2023).

Beyond RNNs, orthogonality or unitarity has also been explored in convolutional and graph models. Unitary CNNs constrain convolutional filters through structured parameterizations or Cayley/Lie-algebra methods (Sedghi et al., 2018; Li et al., 2019b; Singla & Feizi, 2021; Trockman & Kolter, 2021), while Ortho-GConv imposes orthogonality on the feature transformation matrix in GNNs (Guo et al., 2022). Closely related to our work, Kiani et al. (2024) construct unitary graph and group convolutions through matrix exponentials such as $\exp(i\mathbf{A})$, yielding norm-preserving propagation operators with strong empirical performance. GUMP is complementary in both construction and scope: rather than applying a matrix-exponential unitary convolution on the original graph, we transform the input graph into an Eulerian line-graph construction and compute an input-dependent unitary propagation matrix by polar projection while preserving the admissible sparse edge-to-edge transition structure. Orthogonal initialization and optimizers such as Muon further illustrate the usefulness of unitarity at the parameter level (Saxe et al., 2013; Balduzzi et al., 2017; Jordan et al., 2024). More broadly, our setting differs from parameter-level unitarity because the propagation operator is graph-dependent rather than a learned parameter, so enforcing unitarity requires graph transformation and projection rather than only parameter-level constraints.

## B  Preliminaries

**Definition 1** (Line graph). *Given a graph $G$, its line graph $\mathsf{L}(G)$ is a graph such that*

- *each vertex of $\mathsf{L}(G)$ represents an edge of $G$, i.e., $\mathsf{V}[\mathsf{L}(G)] = \mathsf{E}[G]$;*

- *two vertices of $\mathsf{L}(G)$ are adjacent if and only if their corresponding edges share a common endpoint in $G$, i.e., $\mathsf{E}[\mathsf{L}(G)] = \{((i,j),(j,k)) \in \mathsf{V}[\mathsf{L}(G)] \times \mathsf{V}[\mathsf{L}(G)] \mid (i,j),(j,k) \in \mathsf{E}[G]\}$.*

**Definition 2** (Eulerian graph). *An Eulerian graph $G$ is a graph containing an Eulerian cycle, i.e., there is a trail in $G$ that starts and ends on the same vertex and visits every edge exactly once.*

**Definition 3** (Permutation matrix). *A permutation matrix $\mathbf{P} \in \mathbb{R}^{n \times n}$ is a square binary matrix that has exactly one entry of 1 in each row and each column with all other entries 0.*

**Theorem 3.** *Every permutation matrix is orthogonal, i.e., if $\mathbf{P}$ is a permutation matrix, $\mathbf{P}^\top \mathbf{P} = \mathbf{P}\mathbf{P}^\top = \mathbf{I}$.*

## C  Proof

### C.1  Proof of Proposition 1

Proposition 1 is proved based on the following theorem and lemma. Theorem 4 is a direct result from Theorem 3 in Severini (2003). Lemma 1 is a well-known result in graph theory, which can be found in Theorem 1.7.2 of Bang-Jensen & Gutin (2008).

**Theorem 4** (Existence of unitary adjacency matrix). *Let $G$ be a single-connected digraph. Its line graph $\mathsf{L}(G)$ (Definition 1) is the digraph of a unitary matrix if and only if $G$ is Eulerian (Definition 2).*

**Lemma 1** (A special Eulerian graph). *A digraph graph is Eulerian if and only if it is connected and the in-degree and out-degree are equal at each vertex.*

*Proof of Proposition 1.* In Algorithm 1, the undirected edges in $G$ are split into two directed edges in $G'$. Therefore, the in-degree and out-degree of each vertex in $G'$ are equal. We first consider the case where $G$ is connected. Replacing each undirected edge by the pair of directed edges $(i,j)$ and $(j,i)$ preserves connectivity of the underlying graph, so $G'$ is connected as well. By Lemma 1, $G'$ is Eulerian, and Theorem 4 yields a unitary adjacency matrix $\mathbf{U}$ such that $\mathsf{S}[\mathbf{U}] = \mathbf{A}[\mathsf{L}(G')]$.

Now suppose that $G$ is disconnected. Let $G_1, \ldots, G_m$ be the connected components of $G$ that contain at least one edge, and let $G'_1, \ldots, G'_m$ be the corresponding connected components of $G'$ after the edge-splitting step. The same argument as above shows that each $G'_r$ is connected and satisfies $d_{\text{in}}(v) = d_{\text{out}}(v)$ at every

vertex, so Lemma 1 implies that each $G'_r$ is Eulerian. Theorem 4 then gives a unitary adjacency matrix $\mathbf{U}_r$ for each $\mathsf{L}(G'_r)$. Since the line graph of a disjoint union is the disjoint union of the corresponding line graphs, $\mathsf{L}(G')$ is block diagonal up to vertex ordering with blocks $\mathsf{L}(G'_1), \ldots, \mathsf{L}(G'_m)$; isolated vertices of $G$ contribute no vertices to $\mathsf{L}(G')$, but their node features remain available through GUMP's skip/concatenation path. Hence $\mathbf{U} := \mathsf{diag}(\mathbf{U}_1, \ldots, \mathbf{U}_m)$ is unitary and satisfies $\mathsf{S}[\mathbf{U}] = \mathbf{A}[\mathsf{L}(G')]$. $\qquad\square$

## C.2 Proof of Proposition 2

**Lemma 2.** *For any unitary matrix $\mathbf{U}$, given two permutation matrices $\mathbf{P}_1$ and $\mathbf{P}_2$, $\mathbf{P}_1\mathbf{U}\mathbf{P}_2^\top$ is also unitary.*

*Proof.* Let $\hat{\mathbf{U}} = \mathbf{P}_1\mathbf{U}\mathbf{P}_2^\top$. Then, we have

$$\hat{\mathbf{U}}\hat{\mathbf{U}}^\dagger = \mathbf{P}_1\mathbf{U}\mathbf{P}_2^\top\mathbf{P}_2\mathbf{U}^\dagger\mathbf{P}_1^\top = \mathbf{P}_1\mathbf{U}\mathbf{U}^\dagger\mathbf{P}_1^\top = \mathbf{P}_1\mathbf{P}_1^\top = \mathbf{I},$$
$$\hat{\mathbf{U}}^\dagger\hat{\mathbf{U}} = \mathbf{P}_2\mathbf{U}^\dagger\mathbf{P}_1^\top\mathbf{P}_1\mathbf{U}\mathbf{P}_2^\top = \mathbf{P}_2\mathbf{U}^\dagger\mathbf{U}\mathbf{P}_2^\top = \mathbf{P}_2\mathbf{P}_2^\top = \mathbf{I}$$

which proves $\mathbf{P}_1\mathbf{U}\mathbf{P}_2^\top$ is unitary. $\qquad\square$

*Proof of Proposition 2.* Since $\mathbf{A}_w$ is full-rank, the polar factor is given by $\mathsf{U}[\mathbf{A}_w] = \mathbf{A}_w(\mathbf{A}_w^\dagger\mathbf{A}_w)^{-1/2}$. The full-rank assumption is used here to ensure that $\mathbf{A}_w^\dagger\mathbf{A}_w$ is positive definite, so that $(\mathbf{A}_w^\dagger\mathbf{A}_w)^{-1/2}$ exists and the polar factor $\mathbf{A}_w(\mathbf{A}_w^\dagger\mathbf{A}_w)^{-1/2}$ is uniquely defined. Row and column permutations preserve rank, so the same condition holds for $\mathbf{P}_1\mathbf{A}_w\mathbf{P}_2^\top$. Let $\hat{\mathbf{A}}_w = \mathbf{P}_1\mathbf{A}_w\mathbf{P}_2^\top$. Then

$$\hat{\mathbf{A}}_w^\dagger\hat{\mathbf{A}}_w = \mathbf{P}_2\mathbf{A}_w^\dagger\mathbf{P}_1^\top\mathbf{P}_1\mathbf{A}_w\mathbf{P}_2^\top$$
$$= \mathbf{P}_2(\mathbf{A}_w^\dagger\mathbf{A}_w)\mathbf{P}_2^\top.$$

Because permutation similarity preserves matrix functions,

$$(\hat{\mathbf{A}}_w^\dagger\hat{\mathbf{A}}_w)^{-1/2} = \mathbf{P}_2(\mathbf{A}_w^\dagger\mathbf{A}_w)^{-1/2}\mathbf{P}_2^\top.$$

Therefore,

$$\mathsf{U}[\hat{\mathbf{A}}_w] = \hat{\mathbf{A}}_w(\hat{\mathbf{A}}_w^\dagger\hat{\mathbf{A}}_w)^{-1/2}$$
$$= \mathbf{P}_1\mathbf{A}_w\mathbf{P}_2^\top\mathbf{P}_2(\mathbf{A}_w^\dagger\mathbf{A}_w)^{-1/2}\mathbf{P}_2^\top$$
$$= \mathbf{P}_1\mathbf{A}_w(\mathbf{A}_w^\dagger\mathbf{A}_w)^{-1/2}\mathbf{P}_2^\top$$
$$= \mathbf{P}_1\mathsf{U}[\mathbf{A}_w]\mathbf{P}_2^\top,$$

which proves the proposition. $\qquad\square$

## C.3 Proof of Proposition 3

*Proof.* Let $G' = (V, E')$. Index the rows and columns of $\mathbf{A}_w$ and $\mathbf{A}_\mathsf{L}$ by directed edges of $E'$. By Definition 1, for two directed edges $e = (u, v)$ and $f = (x, y)$, the entry $(\mathbf{A}_\mathsf{L})_{ef}$ is one exactly when $v = x$. Thus $(\mathbf{A}_w)_{ef}$ can be nonzero only when $e \in E_{\mathrm{in}}(v)$ and $f \in E_{\mathrm{out}}(v)$ for the same vertex $v \in V$.

Let $\mathbf{P}_{\mathrm{in}}$ be the permutation that groups rows by the sets $E_{\mathrm{in}}(v)$ and let $\mathbf{P}_{\mathrm{out}}$ be the permutation that groups columns by the sets $E_{\mathrm{out}}(v)$. Under these permutations, rows associated with $E_{\mathrm{in}}(v)$ interact only with columns associated with $E_{\mathrm{out}}(v)$, so

$$\mathbf{D} := \mathbf{P}_{\mathrm{in}}\mathbf{A}_w\mathbf{P}_{\mathrm{out}}^\top = \mathsf{diag}\big(\mathbf{B}_v\big)_{v \in V}.$$

Each block $\mathbf{B}_v$ contains all transitions $(u, v) \to (v, w)$ in the line graph. The corresponding block of the binary adjacency $\mathbf{A}_\mathsf{L}$ is the all-ones matrix $\mathbf{J}_v = \mathbf{1}_{d(v) \times d(v)}$, because every incoming directed edge to $v$ can be followed by every outgoing directed edge from $v$ in the line graph. Because Algorithm 1 inserts both orientations of every undirected edge, $|E_{\mathrm{in}}(v)| = |E_{\mathrm{out}}(v)|$, hence $\mathbf{B}_v$ is square.

Since $\mathbf{D}$ is block diagonal,

$$\mathbf{D}^\dagger \mathbf{D} = \mathsf{diag}\big(\mathbf{B}_v^\dagger \mathbf{B}_v\big)_{v \in V}.$$

Matrix functions preserve block diagonality, so under the full-rank assumption,

$$(\mathbf{D}^\dagger \mathbf{D})^{-1/2} = \mathsf{diag}\big((\mathbf{B}_v^\dagger \mathbf{B}_v)^{-1/2}\big)_{v \in V}.$$

Therefore,

$$\begin{aligned}
\mathsf{U}[\mathbf{D}] &= \mathbf{D}(\mathbf{D}^\dagger \mathbf{D})^{-1/2} \\
&= \mathsf{diag}\big(\mathbf{B}_v(\mathbf{B}_v^\dagger \mathbf{B}_v)^{-1/2}\big)_{v \in V} \\
&= \mathsf{diag}\big(\mathsf{U}[\mathbf{B}_v]\big)_{v \in V}.
\end{aligned}$$

By Proposition 2,

$$\mathsf{U}[\mathbf{D}] = \mathsf{U}[\mathbf{P}_{\mathrm{in}} \mathbf{A}_w \mathbf{P}_{\mathrm{out}}^\top] = \mathbf{P}_{\mathrm{in}} \mathsf{U}[\mathbf{A}_w] \mathbf{P}_{\mathrm{out}}^\top.$$

Hence $\mathsf{U}[\mathbf{A}_w] = \mathbf{P}_{\mathrm{in}}^\top \mathsf{U}[\mathbf{D}] \mathbf{P}_{\mathrm{out}}$ has the same row/column block structure as $\mathbf{A}_w$. Since each projected block $\mathsf{U}[\mathbf{B}_v]$ is contained in the corresponding all-ones support block $\mathbf{J}_v$, every nonzero entry of $\mathsf{U}[\mathbf{A}_w]$ lies on an admissible transition of $\mathsf{L}(G')$. Therefore,

$$\mathsf{S}[\mathsf{U}[\mathbf{A}_w]] \subseteq \mathsf{S}[\mathbf{A}_{\mathsf{L}}].$$

If each block polar factor $\mathsf{U}[\mathbf{B}_v]$ is fully supported, this inclusion becomes equality. This condition is generic for full-rank blocks whose entries are produced by continuous learned weights: each entry of $\mathsf{U}[\mathbf{B}_v]$ is a real-analytic function of the entries of $\mathbf{B}_v$ on the full-rank domain, and it is not identically zero because one can choose a fully supported unitary block as $\mathbf{B}_v$. Hence the weights that make a specific projected entry exactly zero form a measure-zero set.

For the Newton-Schulz implementation, let $\mathbf{X}_{v,t}$ denote the $t$-th iterate for block $\mathbf{B}_v$, initialized as $\mathbf{X}_{v,0} = \mathbf{B}_v / \|\mathbf{B}_v\|_F$. If $\mathbf{X}_{v,t}$ is blockwise supported, then $\mathbf{X}_{v,t}^\top \mathbf{X}_{v,t}$ and the polynomial update in Algorithm 2 remain within the same block. By induction, every iterate is block diagonal under the same permutations, so no off-support fill-in is introduced during the iteration either. $\qquad\square$

## C.4 Proofs for Section 3.4

Our proof is based on the GNN model from Figure 2(a) with the activation function being ReLU. GUMP is analyzed with $\mathbf{A}$ being unitary and vanilla message passing is analyzed with $\mathbf{A}$ being the normalized adjacency matrix.

**Proof of Proposition 4.**

*Proof.* Denote by $\mathbf{f}_i^{(l)}$ the pre-activated feature of $\mathbf{h}_i^{(l)}$, i.e., $\mathbf{f}_i^{(l)} = \sum_{z \in \mathcal{N}(i)} \mathbf{A}_{iz} \mathbf{h}_z^{(l-1)} \mathbf{W}_l$, for any $l = 1 \cdots L$, we have

$$\frac{\partial \mathbf{h}_i^{(l)}}{\partial \mathbf{h}_s^{(0)}} = \mathsf{diag}\left(1_{\mathbf{f}_i^{(l)} > 0}\right) \cdot \left( \sum_{z \in \mathcal{N}(i)} \mathbf{A}_{iz} \frac{\partial \mathbf{h}_z^{(l-1)}}{\partial \mathbf{h}_s^{(0)}} \right) \cdot \mathbf{W}_l^\top.$$

By the chain rule, we get

$$\begin{aligned}
\frac{\partial \mathbf{h}_i^{(L)}}{\partial \mathbf{h}_s^{(0)}} &= \sum_{p=1}^{\Psi} \left[ \frac{\partial \mathbf{h}_i^{(L)}}{\partial \mathbf{h}_s^{(0)}} \right]_p \\
&= \sum_{p=1}^{\Psi} \prod_{l=L}^{1} \mathsf{diag}\left(1_{\mathbf{f}_{v_p^l}^{(l)} > 0}\right) \mathbf{A}_{v_p^l v_p^{l-1}} \mathbf{W}_l^\top.
\end{aligned}$$

Here, $\Psi$ is the total number of paths $v_p^L v_p^{L-1} \cdots v_p^1 v_p^0$ of length $L+1$ from $v_p^0 = s$ to $v_p^L = i$. For $l = 1 \cdots L-1$, $v_p^{l-1} \in \mathcal{N}(v_p^l)$.

For each path $p$, the derivative $[\partial \mathbf{h}_i^{(L)} / \partial \mathbf{h}_s^{(0)}]_p$ represents a directed acyclic computation graph. At a layer $l$, we can express an entry of the derivative as

$$\left[ \frac{\partial \mathbf{h}_i^{(L)}}{\partial \mathbf{h}_s^{(0)}} \right]_p^{(m,n)} = \prod_{l=L}^{1} \mathbf{A}_{v_p^l v_p^{l-1}} \sum_{q=1}^{\Phi} Z_q \prod_{l=L}^{1} w_q^{(l)},$$

where $\Phi$ is the number of paths $q$ from the input neurons to the output neuron $(m,n)$, in the computation graph of $[\partial \mathbf{h}_i^{(L)} / \partial \mathbf{h}_s^{(0)}]_p$. For each layer $l$, $w_q^{(l)}$ is the entry of $\mathbf{W}_l^\top$ that is used in the $q$-th path. Finally, $Z_q \in \{0, 1\}$ represents whether the $q$-th path is active ($Z_q = 1$) or not ($Z_q = 0$) as a result of ReLU activation of the entries of $\mathbf{f}_{v_p^l}^{(l)}$'s on the $q$-th path.

Under the assumption that $Z_q$ contributes the path-level gate factor $\rho_L$. Because of $\mathbb{P}[Z_q = 1] = \rho_L, \forall q$, we have

$$\mathbb{E}\left[ \left[ \frac{\partial \mathbf{h}_i^{(L)}}{\partial \mathbf{h}_s^{(0)}} \right]_p^{(m,n)} \right] = \rho_L \prod_{l=L}^{1} \mathbf{A}_{v_p^l v_p^{l-1}} \sum_{q=1}^{\Phi} \prod_{l=L}^{1} w_q^{(l)}.$$

Then, the expected Jacobian is

$$\mathbb{E}\left[ \frac{\partial \mathbf{h}_i^{(L)}}{\partial \mathbf{x}_s} \right] = \sum_{p=1}^{\Psi} \mathbb{E}\left[ \left[ \frac{\partial \mathbf{h}_i^{(L)}}{\partial \mathbf{h}_s^{(0)}} \right]_p \right] = \rho_L \prod_{l=L}^{1} \mathbf{W}_l^\top \left( \mathbf{A}^L \right)_{is}.$$

$\square$

**Average-over-sources identity.** For the following two theorems, let $N$ denote the size of the propagation matrix, fix a target node $i \in [N]$, and sample the source node $s \sim \mathrm{Unif}([N])$. This metric is natural for our analysis because Proposition 4 factorizes the expected Jacobian into the cumulative weight term and the scalar graph-propagation term $(\mathbf{A}^L)_{is}$. Averaging over $s$ therefore measures the influence of a typical source on the fixed target node $i$, while the squared magnitude removes sign or phase cancellations that can make individual entries uninformative. The resulting quantity is exactly the row energy of the propagation matrix at node $i$. With the notation introduced in the main text, this quantity is $I_i(L; \mathbf{P})$.

**Lemma 3.** *For any* $\mathbf{P} \in \mathbb{C}^{N \times N}$ *and any* $L \geq 0$,

$$\mathbb{E}_s\left[ |(\mathbf{P}^L)_{is}|^2 \right] = \frac{1}{N} \sum_{s=1}^{N} |(\mathbf{P}^L)_{is}|^2 = \frac{1}{N} \left\| \mathbf{e}_i^\top \mathbf{P}^L \right\|_2^2 = \frac{1}{N} \left( \mathbf{P}^L (\mathbf{P}^L)^\dagger \right)_{ii}.$$

*Proof.* By definition,

$$\mathbb{E}_s\left[ |(\mathbf{P}^L)_{is}|^2 \right] = \frac{1}{N} \sum_{s=1}^{N} |\mathbf{e}_i^\top \mathbf{P}^L \mathbf{e}_s|^2 = \frac{1}{N} \left\| \mathbf{e}_i^\top \mathbf{P}^L \right\|_2^2.$$

Moreover,

$$\left\| \mathbf{e}_i^\top \mathbf{P}^L \right\|_2^2 = \mathbf{e}_i^\top \mathbf{P}^L (\mathbf{P}^L)^\dagger \mathbf{e}_i = \left( \mathbf{P}^L (\mathbf{P}^L)^\dagger \right)_{ii},$$

which proves the claim. $\square$

**Proof of Theorem 1.**

*Proof.* Since $\mathbf{A}$ is unitary, $\mathbf{A}^L$ is unitary for every $L$, and hence

$$\mathbf{A}^L (\mathbf{A}^L)^\dagger = \mathbf{I}.$$

Applying Lemma 3 with $\mathbf{P} = \mathbf{A}$ gives

$$\mathbb{E}_s\left[\left|(\mathbf{A}^L)_{is}\right|^2\right] = \frac{1}{N}\left(\mathbf{A}^L(\mathbf{A}^L)^\dagger\right)_{ii} = \frac{1}{N}.$$

Since the left-hand side is $I_i(L; \mathbf{A})$, this proves $I_i(L; \mathbf{A}) = 1/N$. □

### C.5   Proof of Theorem 2

*Proof.* Because $\hat{\mathbf{A}}$ is real symmetric, it admits the orthonormal eigendecomposition

$$\hat{\mathbf{A}} = \sum_{j=1}^N \lambda_j v_j v_j^\top, \qquad \hat{\mathbf{A}}^L = \sum_{j=1}^N \lambda_j^L v_j v_j^\top.$$

Since $\mathbf{\Pi} = v_1 v_1^\top$, we also have

$$\hat{\mathbf{A}}^L - \mathbf{\Pi} = \sum_{j=2}^N \lambda_j^L v_j v_j^\top.$$

Applying Lemma 3 with $\mathbf{P} = \hat{\mathbf{A}}$ and using symmetry gives

$$\mathbb{E}_s\left[\left|(\hat{\mathbf{A}}^L)_{is}\right|^2\right] = \frac{1}{N}\left(\hat{\mathbf{A}}^L(\hat{\mathbf{A}}^L)^\top\right)_{ii} = \frac{1}{N}\left(\hat{\mathbf{A}}^{2L}\right)_{ii}.$$

Expanding $\hat{\mathbf{A}}^{2L}$ in the eigenbasis yields

$$\left(\hat{\mathbf{A}}^{2L}\right)_{ii} = \sum_{j=1}^N \lambda_j^{2L} v_{j,i}^2 = v_{1,i}^2 + \sum_{j=2}^N \lambda_j^{2L} v_{j,i}^2,$$

which proves the exact decomposition. Since $|\lambda_j| \le c$ for $j \ge 2$ and $\sum_{j=2}^N v_{j,i}^2 \le 1$,

$$0 \le \frac{1}{N}\sum_{j=2}^N v_{j,i}^2 \lambda_j^{2L} \le \frac{1}{N}c^{2L}\sum_{j=2}^N v_{j,i}^2 \le \frac{1}{N}c^{2L}.$$

For the centered quantity, apply Lemma 3 to $\mathbf{P} = \hat{\mathbf{A}}^L - \mathbf{\Pi}$:

$$\mathbb{E}_s\left[|\delta_L(i,s)|^2\right] = \frac{1}{N}\left((\hat{\mathbf{A}}^L - \mathbf{\Pi})(\hat{\mathbf{A}}^L - \mathbf{\Pi})^\top\right)_{ii}.$$

Using the eigendecomposition above and orthonormality of $\{v_j\}$, we obtain

$$\mathbb{E}_s\left[|\delta_L(i,s)|^2\right] = \frac{1}{N}\sum_{j=2}^N v_{j,i}^2 \lambda_j^{2L} \le \frac{1}{N}c^{2L}.$$

For the lower bound, keep only the terms with $|\lambda_j| = c$:

$$\mathbb{E}_s\left[|\delta_L(i,s)|^2\right] \ge \frac{1}{N}\sum_{j:\,|\lambda_j|=c} v_{j,i}^2 c^{2L} = \frac{1}{N}\alpha_i c^{2L}.$$

Hence the non-stationary source-averaged influence decays at rate $\Theta(c^{2L}/N)$ whenever $\alpha_i > 0$. □

# D   More Experimental Results

**Statistics of datasets**   The statistics of datasets used in experiments are shown in Table 3.

Table 3: Statistics of datasets.

|  | #graphs | Avg. nodes | Avg. edges | Task type |
|---|---|---|---|---|
| Mutag | 188 | 17.9 | 39.6 | Graph Classification |
| Proteins | 1,113 | 39.1 | 145.6 | Graph Classification |
| Enzymes | 600 | 32.6 | 124.3 | Graph Classification |
| NCI1 | 4110 | 29.87 | 32.30 | Graph Classification |
| NCI109 | 4127 | 29.68 | 32.13 | Graph Classification |
| Peptides-func | 15,535 | 150.94 | 307.30 | Graph Classification |
| Peptides-struct | 15,535 | 150.94 | 307.30 | Graph Regression |

**Hyperparameters of GUMP**   The hyperparameters of GUMP for both synthetic and real datasets are shown in Table 4.

Table 4: Hyperparameters of GUMP for datasets in experiments. $\text{layer}_{\text{GUMP}}$, $\text{lr}_{\text{base}}$, $\text{wd}_{\text{base}}$, $\text{lr}_{\text{GUMP}}$, $\text{wd}_{\text{GUMP}}$, drop., $d'$, $d$, batch size, $\text{layer}_{\text{base}}$, opt., sched., and epoch denotes the number of layers of GUMP, the learning rate of base GNN, weight decay of base GNN, the learning rate of GUMP, weight decay of GUMP, dropout rate, dimension of calculating (1), hidden dimension of GNN, batch size, number of layers of base GNN, optimizer, scheduler, and number of epochs, respectively.

|  | $\text{layer}_{\text{GUMP}}$ | $\text{lr}_{\text{base}}$ | $\text{wd}_{\text{base}}$ | $\text{lr}_{\text{GUMP}}$ | $\text{wd}_{\text{GUMP}}$ | drop. | $d'$ | $d$ | batch size | $\text{layer}_{\text{base}}$ | opt. | sched. | epoch |
|---|---|---|---|---|---|---|---|---|---|---|---|---|---|
| CrossedRing | - | $10^{-4}$ | $10^{-6}$ | $10^{-4}$ | 0 | 0 | 32 | 32 | 20 | 0 | adam | none | 200 |
| Ring | - | $10^{-4}$ | $10^{-6}$ | $10^{-4}$ | 0 | 0 | 32 | 32 | 20 | 0 | adam | none | 200 |
| CliquePath | - | $10^{-4}$ | $10^{-6}$ | $10^{-4}$ | 0 | 0 | 32 | 32 | 20 | 0 | adam | none | 200 |
| Mutag | 16 | $10^{-2}$ | $10^{-4}$ | $10^{-4}$ | 0 | 0 | 32 | 64 | 16 | 5 | adam | none | 100 |
| Proteins | 20 | $10^{-2}$ | $10^{-2}$ | $10^{-4}$ | $10^{-2}$ | 0 | 32 | 64 | 64 | 3 | adam | none | 100 |
| Enzymes | 10 | $10^{-2}$ | $10^{-4}$ | $10^{-4}$ | 0 | 0 | 32 | 64 | 16 | 1 | adam | none | 100 |
| NCI1 | 10 | $10^{-2}$ | $10^{-4}$ | $10^{-4}$ | 0 | 0 | 32 | 64 | 16 | 1 | adam | none | 100 |
| NCI109 | 10 | $10^{-2}$ | $10^{-4}$ | $10^{-4}$ | 0 | 0 | 32 | 64 | 16 | 1 | adam | none | 100 |
| Peptides-func | 12 | 0.005 | 0.1 | 0.1 | 0.1 | 0.2 | 32 | 256 | 200 | 3 | adam | cos. | 250 |
| Peptides-struct | 12 | 0.005 | 0.1 | 0.005 | 0.1 | 0.2 | 32 | 256 | 200 | 3 | adam | cos. | 250 |

## D.1   Empirical Block-Rank and Finite-Step Unitarity Diagnostics

**Setup.**   We add diagnostics for the two numerical conditions most directly related to Proposition 2 and Algorithm 2. For each line-graph transition block $\mathbf{B}_v$, we report the full-rank rate under tolerance $10^{-7}$, statistics of the minimum singular value $\sigma_{\min}(\mathbf{B}_v)$, and the finite-step Newton–Schulz residual

$$r_{v,K} = \|\mathbf{X}_{v,K}^{\top}\mathbf{X}_{v,K} - I_{d_v}\|_F/\sqrt{d_v},$$

where $\mathbf{X}_{v,K}$ is the block returned after $K$ Newton–Schulz iterations. The diagnostic uses the same TU line-graph block partition as the implementation and random continuous 64-dimensional node representations passed through the attention form in equation 1, rather than raw discrete TU labels. This avoids degeneracies that can arise from discrete labels and probes the generic continuous-representation regime relevant to learned GUMP blocks.

Table 5: Empirical full-rank and minimum-singular-value statistics for TU transition blocks $\mathbf{B}_v$. Ranks are computed with tolerance $10^{-7}$.

| Dataset | #blocks | max $d_v$ | full-rank (%) | min $\sigma_{\min}$ | p1 $\sigma_{\min}$ | med. $\sigma_{\min}$ |
|---|---|---|---|---|---|---|
| MUTAG | 3,371 | 4 | 100.00 | $2.50\times10^{-6}$ | $1.42\times10^{-4}$ | $1.11\times10^{-1}$ |
| PROTEINS | 43,466 | 25 | 99.99 | $2.30\times10^{-8}$ | $2.07\times10^{-5}$ | $2.65\times10^{-2}$ |
| ENZYMES | 19,474 | 9 | 100.00 | $4.45\times10^{-7}$ | $1.07\times10^{-4}$ | $5.19\times10^{-3}$ |
| NCI1 | 122,319 | 4 | 100.00 | $2.07\times10^{-7}$ | $8.87\times10^{-4}$ | $1.33\times10^{-1}$ |
| NCI109 | 122,020 | 5 | 100.00 | $8.16\times10^{-7}$ | $6.43\times10^{-4}$ | $1.37\times10^{-1}$ |

Table 6: Finite-step Newton–Schulz residuals $r_{v,K} = \|\mathbf{X}_{v,K}^{\top}\mathbf{X}_{v,K} - I\|_F / \sqrt{d_v}$ for $K = 5$ and $K = 10$.

| Dataset | med. $r_{v,5}$ | p95 $r_{v,5}$ | max $r_{v,5}$ | med. $r_{v,10}$ | p95 $r_{v,10}$ | max $r_{v,10}$ |
|---|---|---|---|---|---|---|
| MUTAG | $5.98\times10^{-2}$ | $7.07\times10^{-1}$ | $8.16\times10^{-1}$ | $1.09\times10^{-7}$ | $5.48\times10^{-1}$ | $8.16\times10^{-1}$ |
| PROTEINS | $4.98\times10^{-1}$ | $8.16\times10^{-1}$ | $9.35\times10^{-1}$ | $1.89\times10^{-7}$ | $5.87\times10^{-1}$ | $9.28\times10^{-1}$ |
| ENZYMES | $4.37\times10^{-1}$ | $6.20\times10^{-1}$ | $8.37\times10^{-1}$ | $1.36\times10^{-7}$ | $2.61\times10^{-1}$ | $7.07\times10^{-1}$ |
| NCI1 | $4.86\times10^{-3}$ | $6.84\times10^{-1}$ | $8.64\times10^{-1}$ | $9.78\times10^{-8}$ | $9.34\times10^{-2}$ | $8.09\times10^{-1}$ |
| NCI109 | $5.91\times10^{-3}$ | $6.94\times10^{-1}$ | $8.65\times10^{-1}$ | $9.95\times10^{-8}$ | $2.06\times10^{-1}$ | $8.14\times10^{-1}$ |

These diagnostics support the generic full-rank condition used by Proposition 2: all five datasets have essentially full-rank transition blocks, with only PROTEINS showing a small number of rank-deficient blocks under the conservative $10^{-7}$ tolerance. For finite-step Newton–Schulz projection, increasing from $K = 5$ to $K = 10$ reduces the median normalized residual to approximately $10^{-7}$ across all five datasets. The larger p95 and maximum residuals are concentrated in the ill-conditioned tail of blocks, consistent with the lower tail of $\sigma_{\min}(\mathbf{B}_v)$. Since the projection is blockwise, this tail is localized and reflects the usual trade-off between a fixed iteration budget and numerical accuracy; if a stricter tolerance is desired, it can be addressed by increasing $K$ or selectively applying a higher-accuracy block projection to near-singular blocks.

**Comparison GUMP with more methods**   For the synthetic datasets, we compare GUMP with Drew and ADGN on synthetic datasets in Figure 5. The results show that the performances of GUMP and Drew are close, while ADGN performs worse.

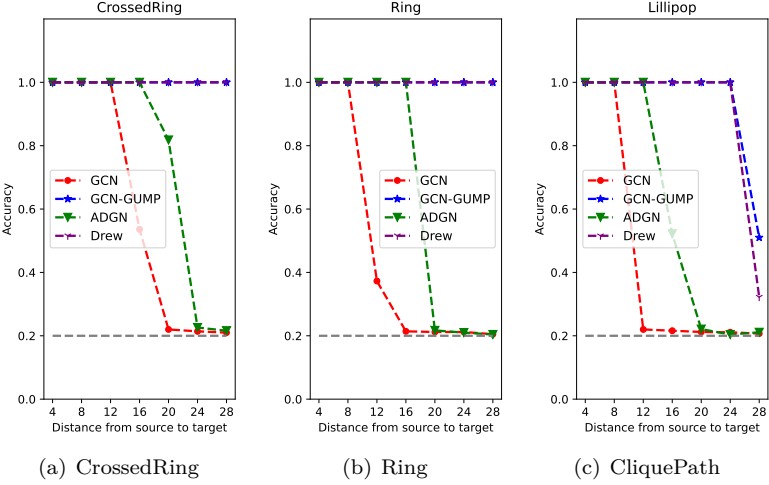

(a) CrossedRing            (b) Ring            (c) CliquePath

Figure 5: The performance of GCN and GCN-GUMP on the CrossedRing, Ring, and CliquePath with different distances from source to target.

We further compare the performances of GUMP by applying different convolution operator. The results are shown in Table 7. The results show that GUMP can improve the performance of both GCN and GIN, while GIN-GUMP achieves better performance than GCN-GUMP.

Table 7: Graph classification accuracy on the TUDataset. **First**, second, and *third* best results are bold, underlined, and underwaved, respectively.

| Base GNN | Methods | Mutag | Proteins | Enzymes | NCI1 | NCI109 | Rank |
|---|---|---|---|---|---|---|---|
| GCN | None | 72.15±2.44 | 70.98±0.74 | 27.67±1.16 | 68.74±0.45 | 67.90±0.50 | 4.2 |
| | None (+layer) | 70.05±1.83 | 69.80±0.99 | 23.63±1.07 | 63.94±1.34 | 55.92±1.26 | 6.8 |
| | DIGL | *79.70±2.15* | 70.76±0.77 | 35.72±1.12 | 69.76±0.42 | 69.37±0.43 | 3.0 |
| | SDRF | 71.05±1.87 | 70.92±0.79 | *28.37±1.17* | 68.21±0.43 | 66.78±0.44 | 4.8 |
| | FoSR | 80.00±1.57 | 73.42±0.81 | 25.07±0.99 | 57.27±0.54 | 56.82±0.60 | 4.6 |
| | GTR | 79.10±1.86 | *72.59±2.48* | 27.52±0.99 | *69.37±0.38* | *67.97±0.47* | 3.6 |
| | GCN-GUMP | **84.89±1.63** | **74.88±0.87** | **36.02±1.43** | **77.97±0.42** | **75.85±0.44** | 1.0 |
| GIN | None | 77.70±3.60 | 70.80±0.83 | 33.80±1.12 | *75.65±0.49* | 74.93±0.46 | 4.0 |
| | None (+layer) | 69.80±2.75 | 68.71±0.96 | 25.92±1.07 | 73.49±0.46 | 72.47±0.53 | 6.6 |
| | DIGL | 79.80±2.08 | 70.71±0.67 | *35.74±1.20* | 79.37±0.43 | 76.88±0.39 | 2.8 |
| | SDRF | *78.40±2.80* | 69.81±0.79 | 35.82±1.09 | 74.55±0.54 | 73.89±0.43 | 4.2 |
| | FoSR | 78.00±2.22 | 75.11±0.82 | 29.20±1.38 | 70.15±0.47 | 69.93±0.45 | 5.2 |
| | GTR | 77.60±2.84 | *73.13±0.69* | 30.57±1.42 | 75.45±0.44 | *75.28±0.42* | 4.2 |
| | GIN-GUMP | **86.72±1.53** | **75.43±0.70** | **48.43±1.24** | **81.25±0.37** | **78.45±0.44** | 1.0 |

We also compare the Jacobian of GUMP with Drew and ADGN in Figure 6. Jacobian of Drew exponentially increases, suggesting its potential numerical instability when training Drew with more layers. The Jacobian of ADGN is small when the ADGN layer is small and steadily increases to $10^{-8}$ as the ADGN layer reaches 100. Although the Jacobian of ADGN does not exhibit exponential decay, the correlation between distant nodes is significantly weaker compared to GUMP.

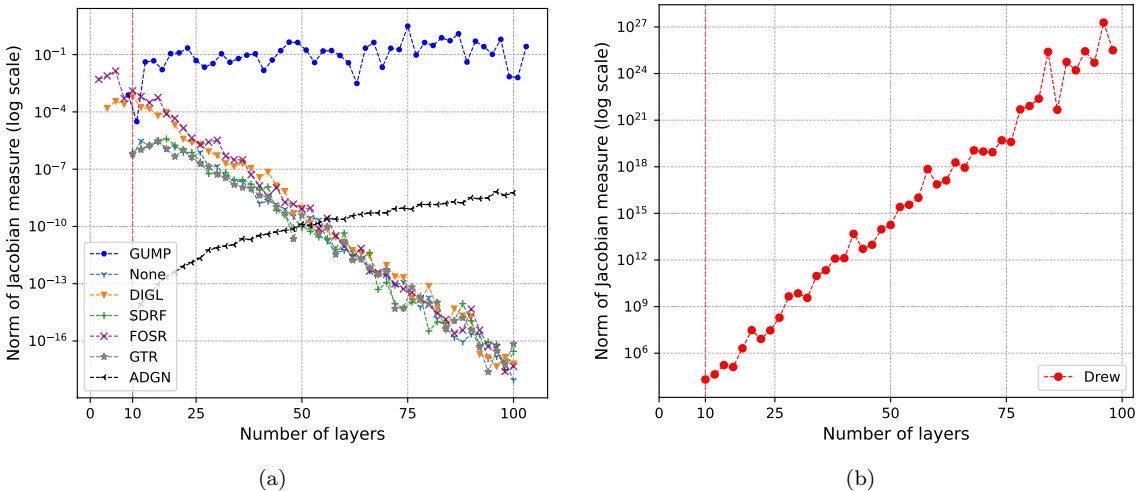

Figure 6: Jacobian norm versus layers on NCI1

## D.2 OGB MOLHIV Results

**Setup** We further evaluate graph classification on `ogbg-molhiv` from the Open Graph Benchmark (OGB) (Hu et al., 2020). The dataset contains molecular graphs and uses scaffold splitting to test generalization to structurally different molecules. Following the official OGB protocol, we report test ROC-AUC.

**Results** Table 8 reports the OGB MOLHIV results with the same comparison methods as Table 1. GIN-GUMP achieves a test ROC-AUC of 77.89, outperforming GRIT at 77.07, GPS at 76.62, GTR at 76.07, and the reproduced Kiani et al. baseline at 75.48. These results provide an additional standardized graph-classification evaluation beyond the TU datasets and show that GUMP remains effective under the official scaffold split.

Table 8: Experimental results on `ogbg-molhiv`. Test ROC-AUC (%) is reported under the official scaffold split and evaluator.

| Classes | Method | Test ROC-AUC (%) ↑ |
|---|---|---|
| – | GCN | 74.47±1.51 |
| Rewiring | DIGL | 75.16±1.81 |
| | SDRF | 75.00±1.08 |
| | FoSR | 73.42±1.35 |
| | GTR | 76.07±0.70 |
| Diffusion | ADGN | 73.21±0.21 |
| | GRAND | 73.29±1.12 |
| Transformer | GPS | 76.62±0.82 |
| – | Ortho-GConv | 70.96±0.64 |
| – | Kiani et al. | 75.48±0.42 |
| K-hop MP | GRIT | 77.07±0.70 |
| Ours | GIN-GUMP | 77.89±0.54 |

### D.3 Node Classification Results

We also apply GUMP to node classification tasks on Cora and Citeseer datasets. The results are shown in Table 9.

Table 9: Accuracy of node classification datasets: Cora and Citeseer

|  | Layers | 2 | 4 | 8 | 16 | 64 |
|---|---|---|---|---|---|---|
| | GCN | 81.1 | 80.4 | 69.5 | 60.3 | 28.7 |
| Cora | GCNII | 82.2 | 82.6 | 84.2 | 84.6 | 85.5 |
| | GCN-GUMP | 84.6 | 86.2 | 84.8 | 85.4 | 87.4 |
| | GCN | 70.8 | 67.6 | 30.2 | 18.3 | 20.0 |
| Citeseer | GCNII | 68.2 | 68.9 | 70.6 | 72.9 | 73.4 |
| | GCN-GUMP | 73.0 | 73.0 | 72.8 | 72.4 | 75.8 |

**Training time of GUMP** The time of training GCN and GCN-GUMP for 100 epochs on the TUDataset is shown in Table 10. Using the same graph-level batch size as GCN is a practical setting for graph classification, but it is not a node-budget-normalized throughput comparison. Since GUMP propagates on the directed line graph, a graph-level mini-batch contains $\sum_g 2|E_g|$ line-graph nodes rather than $\sum_g |V_g|$ original graph nodes. Therefore, we also report a node-budget-matched GUMP setting with

$$B_{\text{GUMP}}^{\text{match}} = \max\left(1, \left\lfloor B_{\text{GCN}} \frac{\bar{n}}{2\bar{m}} \right\rfloor\right),$$

where $\bar{n}$ and $\bar{m}$ are the average numbers of nodes and edges per graph. This setting controls the expected number of propagated nodes per mini-batch, making the comparison fairer in terms of the amount of graph state propagated in each update.

Table 10: Training seconds on TUDataset for 100 epochs. The same-batch setting uses the shared GCN/GUMP graph-level batch sizes from the accuracy experiments, while the matched setting controls the expected propagated-node budget per mini-batch.

| Dataset | GCN time (s) | GCN/GUMP graph batch | Same-batch GUMP time (s) | Matched GUMP batch | Matched GUMP time (s) |
|---|---|---|---|---|---|
| MUTAG | 4.39 | 16 | 7.21 | 3 | 37.9 |
| Proteins | 20.57 | 64 | 26.19 | 8 | 203.7 |
| Enzymes | 11.26 | 16 | 20.29 | 2 | 160.2 |
| NCI1 | 71.79 | 16 | 82.02 | 7 | 187.7 |
| NCI109 | 74.56 | 16 | 93.38 | 7 | 213.5 |

Under this matched propagated-node budget, GUMP is more expensive than GCN: the matched GUMP times on MUTAG/PROTEINS/ENZYMES/NCI1/NCI109 are 37.9/203.7/160.2/187.7/213.5 seconds, respectively, which are larger than the corresponding GCN times. Thus, Table 10 should be interpreted as showing the practical same-batch runtime and the additional compute cost incurred by GUMP under a fairer propagated-node budget, rather than as evidence of comparable per-node cost to GCN.

# E   Apply GUMP to Directed Graphs

GUMP can also be applied to directed graphs as shown in Algorithm 4. The transformation of directed graphs is also based on Lemma 1.

---
**Algorithm 4** Graph transformation for directed graphs

---
**Require:** A directed graph $G = (V, E)$;
 1: Initialize a new digraph $G' = (V, E')$;
 2: **for** $(i, j) \in E$ **do**
 3:     Add $(i, j)$ and $(j, i)$ to $E'$;
 4: **end for**
 5: Remove duplicated edges in $E'$;
 6: Convert $G'$ to its line graph $\mathsf{L}(G')$;
 7: **Return:** A digraph $\mathsf{L}(G')$.

---

