# OpenReview forum: "Graph Unitary Message Passing"
_TMLR — Accepted by TMLR_

### Review · Reviewer_4fXB · 2026-02-17

**Summary Of Contributions:**

**Summary**

The paper aims to address the over-smoothing problems in GNNs by introducing Graph Unitary Message Passing (GUMP). Drawing inspiration from Unitary RNNs, the authors propose replacing standard weight matrices with unitary adjacency matrices to preserve signal norms during deep message passing. The method involves transforming an input graph into the line graph of its Eulerian extension, which is a structure theoretically guaranteed to admit a unitary adjacency matrix. The authors then learn edge weights and project the resulting weighted adjacency matrix onto the unitary manifold using Newton-Schulz iterations. Theoretical analysis suggests this approach bounds the Jacobian norm, stabilizing training for deep networks. Experiments on synthetic tasks, TUDataset, and LRGB benchmarks report performance improvements over various baselines.

**Strengths**
- The paper draws a compelling parallel between the vanishing gradient problem in RNNs and over-smoothing in GNNs. Leveraging the success of Unitary RNNs and applying it to graph propagation is a logical and interesting research direction.
- The application of Newton-Schulz iterations to graph adjacency matrices is a novel technical integration in the GNN context.
- Empirical results cover multiple task types and include long-range settings.

**Weaknesses**
- The central premise of the paper is the use of a "Unitary Adjacency Matrix". An adjacency matrix implies sparsity defined by the graph structure. However, the Polar Decomposition used in Lemma 1 finds the closest unitary matrix in the entire manifold of unitary matrices. The polar factor (and the result of Newton-Schulz iterations) of a sparse matrix is almost strictly a dense matrix. The paper never proves (or even states) a condition under which the polar factor would preserve the original support. Algorithm 3 then computes messages using a summation $\sum_{u \in N_v} U[A_w]_{vu}$. This implies the method relies on the sparsity of the original line graph. If $U[A_w]$ is dense, but Algorithm 3 only uses entries corresponding to edges $E$, the effective operator is a truncated/masked version of $U[A_w]$. A truncated unitary matrix is not unitary. This voids the premise of "Unitary Message Passing" and the applicability of Theorem 1.
- The Newton-Schulz iteration involves matrix-matrix multiplication of the adjacency matrix $X_t$. If $X_t$ becomes dense or even if it remains sparse but fills in, calculating $X_t (X_t^\top X_t)$ is extremely expensive compared to sparse matrix-vector multiplication used in standard GNNs. There is no complexity analysis regarding the number of edges. For a graph with many edges, the $O(E^3)$ or even $O(E^{2.37})$ complexity of dense matrix multiplication required for the projection step seems prohibitive.
- Why are the axes in Figure 1 curly?

**Audience:**

Yes

**Audience Explanation:**

The community interested in GDL and GNN expressivity would find the connection to Unitary RNNs and the use of line graphs interesting.

**Claims And Evidence:**

No

**Claims Explanation:**

Claims about empirical performance improvements are supported by the provided results on multiple benchmarks. Claims that Jacobian behavior is more stable with depth are supported by the plotted Jacobian norms.

The manuscript’s definition of “Unitary Adjacency Matrix” requires support preservation, but the constructive method used does not establish that the resulting unitary keeps the same sparsity pattern. The implementation effectively masks this matrix to the graph support but provides no evidence (theoretical or empirical) that the masked projection retains unitarity.

**Requested Changes:**

- Make the support/connectivity story rigorous by either proving that the computed projection$U[A_w]$ preserves the sparsity pattern required by the definition $S[U_G] = S[A]$, or redefine the “unitary adjacency matrix” concept used in the method to match what Algorithm 2 actually outputs.
- Provide a Big-O complexity analysis for the Graph Transformation and the Newton-Schulz iteration in terms of $|V|$ and $|E|$. Discuss the scalability implications for large graphs.
- Fix Figure 1.

---

### Review · Reviewer_guc8 · 2026-02-18

**Summary Of Contributions:**

This paper addresses the oversquashing/oversmoothing issue of message passing in GNNs by proposing the GUMP algorithm, which essentially relies on the construction of unitary adjacency matrices. A proof of the existence of unitary adjacency matrices and an efficient algorithm for approximating such matrices are provided. The primary theoretical result shows that under certain assumptions, the expected entries of the Jacobian (of the hidden states with respect to the input) are $O(1)$ under GUMP but $O(c^L)$ under vanilla message passing for $c\in(0,1)$ and $L$ being the number of layers. Empirical results support the theoretical findings and demonstrate improvements on several graph learning tasks.

**Additional Comments:**

1. I have carefully checked the proofs and they are correct apart from the previously mentioned points.
2. It has since been shown that Muon benefits from more accurate and efficient approximations of the polar factor than the Newton-Schulz iteration provided by [JJB+24], e.g. [APMG26]. Are these improvements similarly reflected in GUMP?
3. I am not familiar with graph learning tasks and leave it to the other reviews to comment on the significance of the empirical results.

References
[JJB+24] Keller Jordan, Yuchen Jin, Vlado Boza, Jiacheng You, Franz Cesista, Laker Newhouse, Jeremy Bernstein. Muon: An optimizer for hidden layers in neural networks. 2024.
[APMG26] Noah Amsel, David Persson, Christopher Musco, Robert M. Gower. The Polar Express: Optimal Matrix Sign Methods and Their Application to the Muon Algorithm. ICLR 2026.

**Audience:**

Yes

**Audience Explanation:**

GNNs are an important paradigm in machine learning, and this paper would be of interest to individuals working in that area.

**Claims And Evidence:**

Yes

**Claims Explanation:**

The main algorithm is clearly stated and supported by theoretical and empirical justification.

**Requested Changes:**

1. In the proof of Theorem 4, the authors assume that all paths in the computation graph of the GNN are activated with the same probability $\rho$. Can the authors elaborate on why this is a reasonable assumption? It would be good to provide additional theoretical or empirical evidence.
2. It seems like both $\mathbf{W}$ and $\mathbf{A}$ need to be unitary or approximately unitary in order for the desired results to hold. The authors should explicitly clarify how the $\mathbf{W}$ matrices are maintained.
3. The desired result of Theorem 1 is that under GUMP, the expected entries of the Jacobian do not asymptotically vanish as with vanilla message passing. However, an $O(1)$ bound is provided instead of the expected $\Omega(1)$. Please justify this.
4. The manuscript could greatly benefit from additional proofreading and revision. There are many grammatical errors (e.g. in the abstract and Sections 1 and 2), and there is a lot of redundant material between Sections 2 and 3.
5. (minor) The display on page 2 and line 6 of Algorithm 3 use conflicting notation. What is $\gamma$?
6. (minor) Towards the end of the proof of Lemma 1, it should read "$\mathbf{M}^\dagger(\mathbf{A}_w^\dagger\mathbf{A}_w)^\frac{1}{2}\mathbf{M}$ is positive semi-definite" instead.
7. (minor) Proposition 2 is perhaps more clearly proven by directly computing $\mathsf{U}[\mathbf{P}_1\mathbf{A}_w\mathbf{P}_2^\top]=\mathbf{P}_1\mathbf{A}_w\mathbf{P}_2^\top((\mathbf{P}_1\mathbf{A}_w\mathbf{P}_2^\top)^\dagger\mathbf{P}_1\mathbf{A}_w\mathbf{P}_2^\top)^{-\frac{1}{2}}=\mathbf{P}_1\mathbf{A}_w(\mathbf{A}_w^\dagger\mathbf{A}_w)^{-\frac{1}{2}}\mathbf{P}_2^\top=\mathbf{P}_1\mathsf{U}[\mathbf{A}_w]\mathbf{P}_2^\top$.
8. (minor) For the proof of Theorem 1, it should be noted that if $\mathbf{A}$ is unitary, then $\mathbf{A}^L$ is unitary, and unitary matrices have entries with absolute value bounded by 1. The proof of Theorem 5 is also somewhat handwavy. It would be better to provide a specific $\mathbf{A}$ achieving the bound.

---

### Review · Reviewer_n2fn · 2026-04-16

**Summary Of Contributions:**

GUMP is a graph learning procedure which transforms a graph to its line graph and learns on it a unitary adjacency matrix via Newton-Schulz iteration for message passing in the GNN. The authors  show theoretically how this contributes to overcoming oversmoothing. They provide synthetic experiments and empirical validation on graph classification tasks.

**Strenghts**

1)  Unitary constraints have gained interest in other ML fields, but adapting them to the message passing scheme rather than the weight matrix directly accounts for the structural constraints of graph data.

2) The use of line graphs to enable unitary adjacency matrices is an original idea.

3) The synthetic experiments are convincing, showing that GUMP improves long-range information propagation. The norm of the Jacobian analysis is also a good insight.

**Main Weaknesses**


1) Severini's result is invoked in Proposition 1 to justify GUMP construction on a line graph, but Severini only guarantees the existence of a unitary matrix with the correct sparsity pattern for the line graph. It is not clear that $U[A_w]$ actually inherits this support, since the polar projection minimizes over all unitary matrices, most of which do not have the correct sparsity pattern. Is support preservation proved theoretically or observed empirically ?


2) The paper does not discuss Kiani et al. "Unitary convolutions for learning on graphs and groups" (NeurIPS 2024), which is a closely related approach. They propose a unitary propagation matrix via the exponential map $exp(iA)$ unitary by construction with formal guarantees and strong empirical results on heterophilic node classification, LRGB benchmarks, and even with TUD datasets in appendix. A positioning and empirical comparison with this work is necessary.


3) The paper includes a limitations section but does not mention computational and memory complexity, which in my view is the main bottleneck of the method and likely explains why experiments are restricted to graph classification on small graphs rather than node classification on larger ones.
For a graph with $N$ nodes and $E$ edges and networks with hidden dimension $D$, a standard GCN runs in $\mathcal{O}(ED+ ND^2)$.
GUMP operates on the line graph that has $2E$ nodes, as it is indicated in the manuscript, but also  $\sum_{i=1}^N d_i^2/2 -E$ edges which is not mentioned in the paper. The time complexity of the GCN on the line graph is thus $\mathcal{O}((\sum_{i=1}^N d_i^2/2 -m)D+ 2ED^2)$.
This does not even account for the computation of the learned edge weights of $A_w$ and the Newton Schulz iterations, which add further overhead. Complexity should be analyzed theoretically on concrete examples such as degree-regular graphs and empirical runtime and memory usage across larger graphs should also be reported.

4) Table 7 in appendix reports training times for GCN and GUMP but does not specify whether the same batch size was used for both. If so, the comparison is unfair, as the line graph has $2E$ nodes, meaning that for a given batch size GUMP processes significantly more nodes per batch than GCN and thus benefits from higher GPU utilization. For instance, in MUTAG where $E/N \approx 2$ a fair comparison would require using a batch size 4 times smaller for GUMP to match the same total number of nodes per batch. Without this adjustment, the reported training times likely underestimate the true computational cost of GUMP-GCN relative to GCN on the original graph.

**Minor Weaknesses**

1) Lemma 1 and Proposition 2 follow directly from Keller (1975), already cited in the paper, who shows that the unitary polar factor uniquely minimizes the Frobenius distance to a given matrix over all unitary matrices. When $A_w$ the polar decomposition admits the closed form proposed. In my opinion, separate proofs are not needed and risk obscuring a known result.


2) Proposition 1 builds on Severini's result which requires the graph to be connected, but this assumption is not explicitly stated in the main manuscript. In practice, Cora, used in the appendix experiments, is not fully connected in its default form, and it is unclear how the proposed method handles disconnected graphs for their protocol.

3) It is unclear if the performance gains come from the line graph transformation or from the learned unitary projection. A GCN run directly on the line graph without learned edge weights is a missing ablation.

4) The paper does not discuss existing work on line graph neural networks, which are mostly designed for link prediction but remain closely related to the proposed graph transformation.

5) Several results in the appendix, including node classification experiments and GIN comparisons, are not referenced in the main text.

6) Results on TUDatasets such as MUTAG are inconsistent across the literature, see values reported in the original GIN paper for GCN, suggesting high sensitivity to experimental setup. Datasets such as MOLHIV or MOLBBBP from OGB are now more widely adopted by the community for graph classification.

**Audience:**

Yes

**Audience Explanation:**

The idea of using unitary matrices for message passing via line graphs is novel and likely to interest the graph machine learning community. Unitarity has proven effective in other areas but remains underexplored for graph data. That said, due to the computational overhead, the method in its current form may only be practical for small graph classification datasets.

**Broader Impact Concerns:**

None identified

**Claims And Evidence:**

No

**Claims Explanation:**

Some theoretical claims are not fully supported, as discussed in the weaknesses. On the empirical side, the synthetic experiments are convincing but the absence of any computational cost analysis makes it hard to assess whether GUMP is a practical solution beyond small graph classification tasks. The omission of Kiani et al. (NeurIPS 2024) also makes it difficult to situate the contribution.

**Requested Changes:**

Address the main weaknesses raised above: compare empirically and theoretically with Kiani et al., analyze the computational and memory cost of the method, and clarify if support preservation of $U[A_w]$ is theoretically guaranteed. Experiments on larger datasets such as MOLHIV or MOLBBBP from OGB with reported runtime and memory would strengthen the evaluation.

---

### Decision · Action_Editor_pr9k · 2026-06-01

**Recommendation:** Accept with minor revision

**Additional Comments:**

Changes to be included in the revised version are:
- a clearer mention of limited scalability, as highlighted by all reviewers
- an improved presentation of Theorem 1 (rev `n2fn`: the objects and notation introduced are non-standard, and the result should connect more clearly to the existing oversmoothing literature through the Jacobian and the role of the eigenvalues of the propagation matrix.)

**Audience:**

Yes

**Audience Explanation:**

Oversmoothing is a widely known problem, using unitary adjacency matrix as a remedy is a novel idea. In particular, using such constraints is the message passing scheme is an original idea, that is adequately investigated in the submission.

**Claims And Evidence:**

Yes

**Claims Explanation:**

The paper proposes Graph Unitary Message Passing, an algorithm that transforms a graph into its line graph and relies on Newton-Schulz iterations to learn a unitary adjacency matrix for message passing in the GNN. Using unitary adjacency matrices is claimed to reduce oversmoothing.

Reviewers had initial criticism towards the paper; some proofs were clarified, the presentation was improved and additional results (e.g. block rank required by rev `4fXB`, tests on OGB MOLHIV required by rev `n2fn`). All three reviewers now recommend acceptance.
The contribution is sound and the empirical validation satisfactory (e.g. providing results similar to the work of Kiani et al pointed out by rev `n2fn`); it is to be noted that in its current form the method remains limited to small graphs.